# Supramolecular aptamer nano-constructs for receptor-mediated targeting and light-triggered release of chemotherapeutics into cancer cells

Deepak K. Prusty[1,2], Volker Adam[1], Reza M. Zadegan [3], Stephan Irsen[4] & Michael Famulok [1,2]

Platforms for targeted drug-delivery must simultaneously exhibit serum stability, efficient directed cell internalization, and triggered drug release. Here, using lipid-mediated self-assembly of aptamers, we combine multiple structural motifs into a single nanoconstruct that targets hepatocyte growth factor receptor (cMet). The nanocarrier consists of lipidated versions of a cMet-binding aptamer and a separate lipidated GC-rich DNA hairpin motif loaded with intercalated doxorubicin. Multiple 2′,6′-dimethylazobenzene moieties are incorporated into the doxorubicin-binding motif to trigger the release of the chemotherapeutics by photoisomerization. The lipidated DNA scaffolds self-assemble into spherical hybrid-nanoconstructs that specifically bind cMet. The combined features of the nanocarriers increase serum nuclease resistance, favor their import into cells presumably mediated by endocytosis, and allow selective photo-release of the chemotherapeutic into the targeted cells. cMet-expressing H1838 tumor cells specifically internalize drug-loaded nanoconstructs, and subsequent UV exposure enhances cell mortality. This modular approach thus paves the way for novel classes of powerful aptamer-based therapeutics.

[1] Life and Medical Sciences (LIMES) Institute, Chemical Biology & Medicinal Chemistry Unit, c/o Kekulé Institute of Organic Chemistry and Biochemistry, Gerhard-Domagk-Strasse 1, 53121 Bonn, Germany. [2] Stiftung Caesar, Max-Planck-Fellowship Group Chemical Biology, Ludwig-Erhard-Allee 2, 53175 Bonn, Germany. [3] Nanoscale Materials & Device Group, Micron School of Materials Science and Engineering, Boise State University, Boise, USA. [4] Stiftung Caesar, Elektronenmikroskopie und Analytik, Ludwig-Erhard-Allee 2, 53175 Bonn, Germany. Correspondence and requests for materials should be addressed to M.F. (email: m.famulok@uni-bonn.de)

There is a compelling demand for improvements in the effectiveness in both the transport and specific release of therapeutic molecules. A powerful approach is the use of aptamer-based tumor targeting systems[1–5] in combination with controlled release of active therapeutics through physico-chemical responses to external stimuli such as pH[6–9], light[10–12], and chemicals[13–15], or internal cell markers[16,17]. Due to their advantages over other targeting reagents such as easy synthesis, low immunogenicity, and high target affinity, DNA aptamers have opened up new opportunities for cellular targeting and have been selected against various cancer types, including prostate[18–20], pancreatic[21,22], colon[23,24], and breast cancer[25–27]. However, aptameric molecular nanocarriers are often limited by inefficient cellular uptake and short intracellular half-life as they are naturally susceptible to nuclease-mediated degradation.

Progress has been made to improve serum half-life and cell internalization efficacy by functionalizing nanocarriers with aptamers that target specific surface proteins, for instance polymeric nanoparticles[28,29], liposomes[30–33], aptamer-drug conjugates[34–36], aptamer-antibody conjugates[37,38], and aptamer-functionalized quantum dots[39–41]. However, the majority of these approaches entailed significant trade-offs between complicated assembly, suboptimal size, limited payload capacity, and some show insufficient serum stability and cell internalization efficacy. In the case of aptamer-drug conjugates, covalent linking of targeting units to cytotoxic agents is one possibility for efficient treatment; however, in some cases limited by the concern that the attachment may alter their biological activity. Several recent studies employed a native cell-targeting aptamer that was modified by additional nucleobases for drug intercalation as a dual

factor for cell targeting and, simultaneously, as a cargo for drug transport[42–44]. Yet, there is an inherent limitation to broader applicability for such architectures: especially when extended to other aptameric platforms for targeting different cell types, even a minor modification of the aptamer sequence with a drug loading unit might result in significant disruption of binding affinity.

An alternative and highly versatile approach to minimize these drawbacks is to incorporate a cell-targeting aptamer unit and separate drug-carrying functionalities into a single multi-functional nano-assembly. These units can be anchored onto a single nanoscaffold through non-covalent interactions, enabling convenient self-assembly of tunable modular components. The advantage of such a system is that simple mixing of the two, or more, moieties would spontaneously self-assemble into a single nanoconstruct containing these motifs. A possible strategy to explore this concept would be harnessing the lipid-based self-assembly of two lipidated structures, one for cell-targeting, the other for drug loading. Potentially suitable candidate cell-targeting moieties are DNA aptamers that bind to extracellular domains of transmembrane receptors, an example being the DNA aptamer cln003[45], which binds with high specificity and affinity to the transmembrane receptor "hepatocyte growth factor receptor" HGFR (also called cMet)[46]. cMet is expressed on the surface of numerous solid tumors.

The DNA-intercalating drug doxorubicin (**DxR**) is one of the most potent and widely used chemotherapeutics, but its lack of specificity induces adverse side effects and toxicities. Substantial efforts in transforming the use of free **DxR** into targeted **DxR**-carrier systems were undertaken[47–49], but a common limitation is inefficient drug release. Designing multi-functional nano-

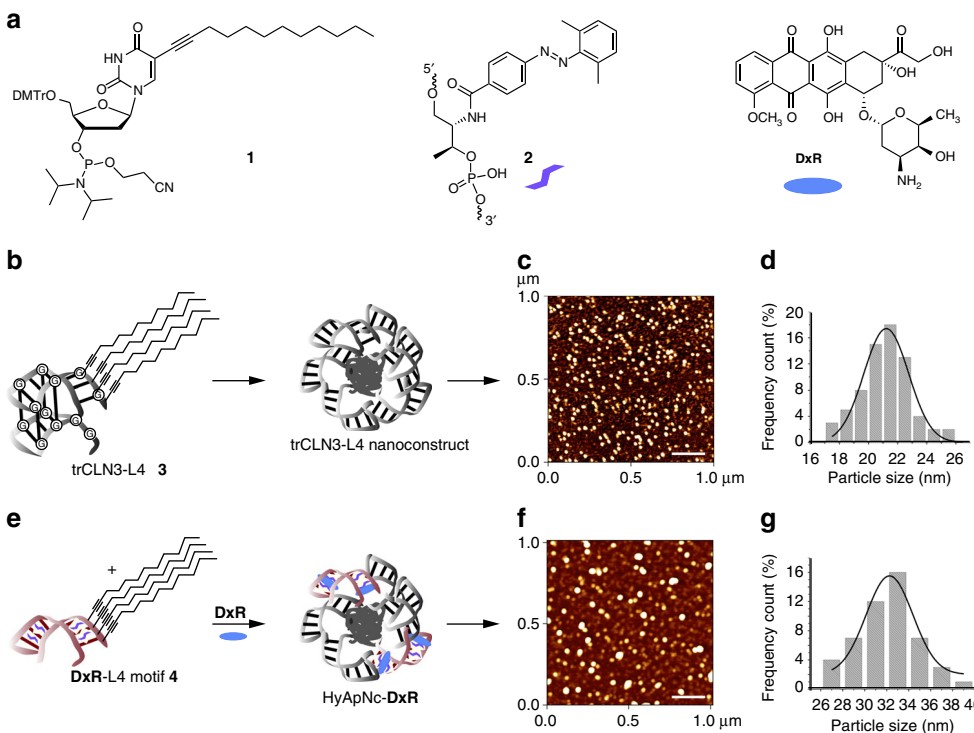

**Fig. 1** Assembly and characterization of the photo-switchable aptameric nanoconstruct. **a** Structures of the lipid-functionalized dU-phosphoramidite **1**, the 2′,6′-dimethylazobenzene-D-threoninol residue **2**, and doxorubicin **DxR**. **b** The lipid-functionalized anti-cMet aptamer trCLN3-L4 **3** and its self-assembly into the corresponding trCLN3-L4 nanoconstruct. **c** AFM images of the trCLN3-L4 **3** nanoconstruct shows its size and morphology. Scale bar: 200 nm. **d** Size distribution of the trCLN3-L4 **3** nanoconstructs. **e** The lipid-functionalized **DxR**-carrier hairpin motif **DxR**-L4 **4** modified with 2′,6′-dimethylazobenzene **2** (purple), and the self-assembly of **3**, **4**, and **DxR** (blue) to form **DxR**-loaded HyApNc-**DxR** nanoconstruct. **f** AFM images of the HyApNc-**DxR** nanoconstruct shows its size and morphology. Scale bar: 200 nm. **g** Size distribution of the HyApNc-**DxR** nanoconstructs. The size distributions in **d** and **g** show that the hybrid nanoconstructs HyApNc-**DxR** are on average about 10 nm larger than the homogeneous trCLN3-L4 nanoconstructs

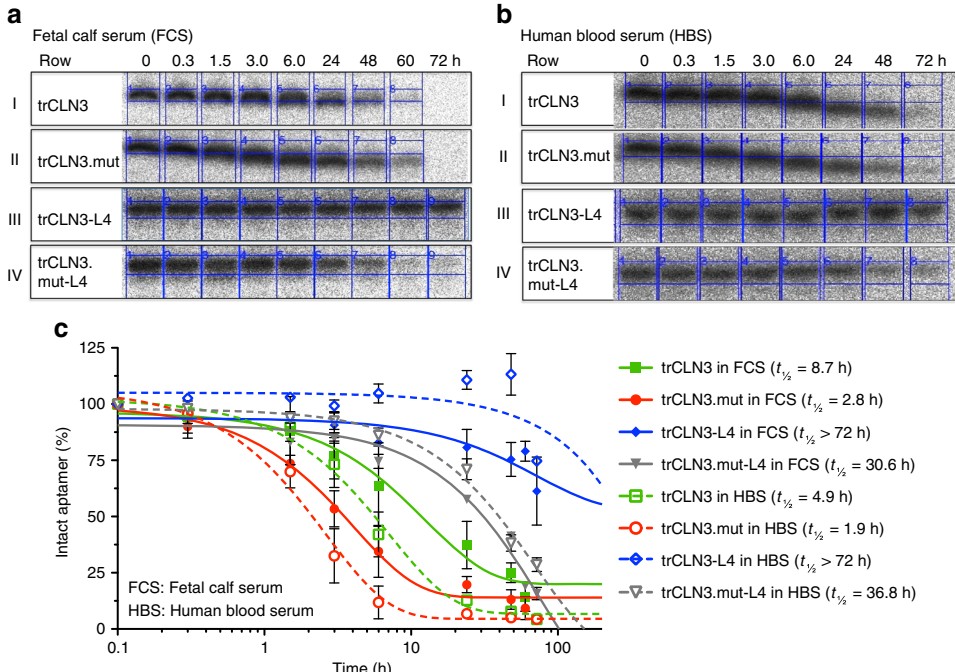

**Fig. 2** Serum stability of trCLN3 aptamer and its lipid-functionalized derivatives. **a, b** PAGE-analysis of the degradation pattern of aptamer trCLN3, trCLN3. mut, trCLN3-L4, and trCLN3.mut-L4 in 90% PBS-buffered fetal calf serum (FCS) and 90% PBS-buffered human blood serum (HBS), respectively. $\gamma$-$^{32}$P-ATP-labeled aptamer bands of the unmodified trCLN3 (row I), trCLN3.mut (row II), trCLN3-L4 (row III), and trCLN3.mut-L4 (row IV), respectively, at different time intervals. Bands at the migration level of the 0 h sample represent 100% intact aptamer, whereas signals at lower positions correspond to decomposition products. **c** Comparison of the degradation pattern of lipidated vs. non-lipidated motifs at different time point of 0.3–72 h. Aptamer band intensities were calculated from gels as in **a, b**, row I–IV; the percentage of intact aptamer was calculated and a curve was fitted to the resulting time course. The half-lives ($t_{1/2}$) of the selected aptamers were determined from the half-life curve fitting and are shown in brackets of the corresponding legends ($n = 2$, error bars: mean $\pm$ S.D.)

constructs as delivery vehicles without an efficient release mechanism will probably limit the development of a potent drug delivery platform. The often-used antisense-strategy[50,51] is both highly complex and polluting, due to the resulting ODN "waste" strands, hence limited as an efficient release system. In contrast, light is an excellent tool for both actively and remotely controlling the release of biologically active caged compounds[52,53]. Systematic investigation of light for cytotoxic drug release is scarce, though it promises simple active control with minimal waste accumulation and is well suited for ODN-based carrier systems. Photoresponsive azobenzene derivatives have been incorporated into ODN-backbones to reversibly open and close ODN-duplexes upon light irradiation[54,55]. Such light-responsive systems can control cellular[56,57] and biological activities like regulation of gene expression[58] or reversible tuning of DNA nanoarchitectures[59–61]. However, studies that systematically explore the control of cytotoxic drug release from aptameric cages modified with photo-responsive moieties inside the ODN-backbone are scarce.

Here, we report the design of a versatile and broadly applicable photo-switchable hybrid-aptameric nanoconstruct (HyApNc) as an efficient molecular carrier that selectively targets and transports high doses of **DxR** directly into cMet-expressing cells and releases the payload under light irradiation. Specifically, we use a truncated, variant of cln003 comprising 40 nucleotides (trCLN3) that binds to cMet with nanomolar affinity and was previously used for aptamer-based affinity labeling of cMet in living cells[62]. For the drug-carrying domain, a separate 5′-GC-rich hairpin ODN motif was designed to intercalate and transport **DxR**. We incorporate 2′,6′-dimethylazobenzene (DMAB) photoswitches into the drug-carrying ODN domain, enabling controlled release

of **DxR** through UV irradiation. Our concept of a photo-switchable HyApNc addresses key concerns for aptameric delivery platforms: specificity, stability, tuneability, and triggered release. The exemplary system we introduce selectively targets cMet-expressing cells with the aptamer trCLN3 and transports high doses of **DxR** for release under irradiation. These active components (both trCLN3 and **DxR**-binding motifs) are chemically modified with four $C_{12}$-lipids (L4), enabling self-assembly into rigid, multi-component supramolecular structures. The aptamer-motif trCLN3-L4 (**3**) allows specific binding to cMet on the cell surface, leading to internalization of the HyApNc, carrying **DxR** intercalated into the **DxR**-L4 motif. Photo-isomerization of DMAB triggers de-hybridization of the double helix and release of **DxR**. The design of the system allows using a large variety of other types of lipid-modified aptamers or molecules that all can, in principle, self-assemble into a single functional nanoconstruct so that a highly versatile applicability of this platform becomes possible.

## Results

**Assembly of anti-cMet trCLN3-L4 nanoconstructs.** As a proof of concept, we used the 40-nucleotide anti-cMet DNA aptamer trCLN3 that binds to HGFR (cMet) with a dissociation constant ($K_d$) of 38 nM[62]. cMet is expressed on the surface of several types of cancer cells, including the NCI-H1838 lung cancer cell line used here[63,64]. In a first step (Fig. 1), we synthesized the lipid-modified phosphoramidite **1** with a $C_{12}$-lipid chain incorporated at the 5-position of the uridine base (Fig. 1a; Supplementary Figs. 1, 2). Four of these modified bases were attached to the 5′-end of the trCLN3 aptamer (Fig. 1b; Supplementary Fig. 3),

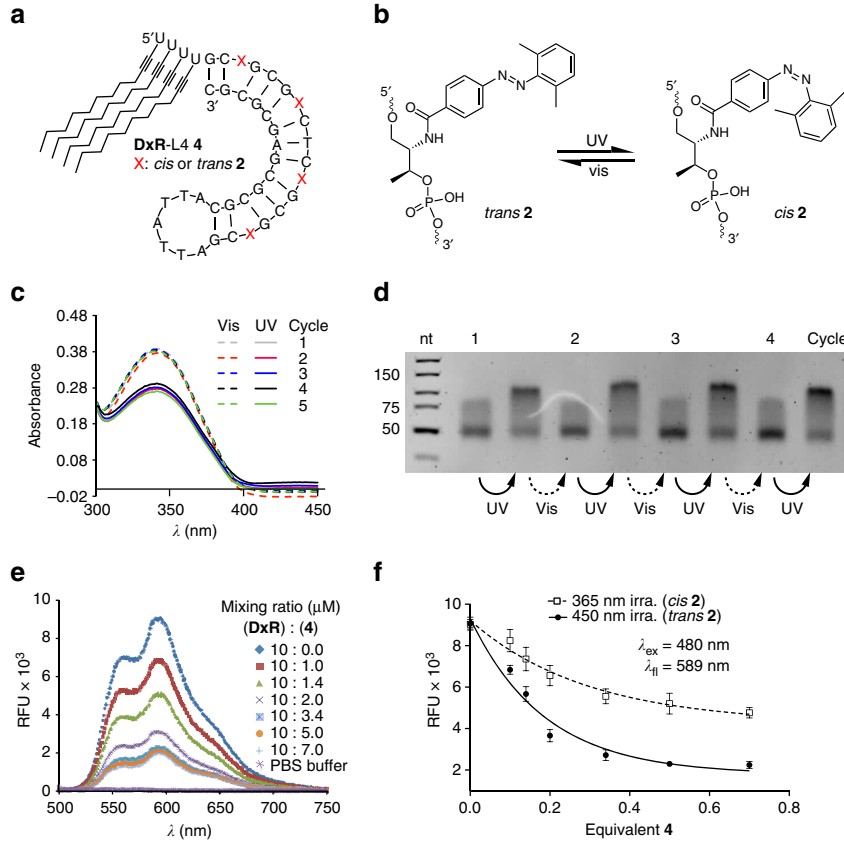

**Fig. 3** Switching behavior of the **DxR**-binding motif. **a** Schematic of lipid-modified hairpin-duplex motif with repetitive 5′-CG-3′ base pairs for **DxR** intercalation. The modified **DxR**-L4 motif **4** shows the positions of DMAB switches on a D-threoninol backbone marked with a red cross X = 2′,6′-dimethylazobenzene; and four lipid chains are attached to the 5′-end. **b** Schematic of the switch mechanism mediated by the DMAB photoswitch. **c** UV/vis spectrum of **DxR**-L4 motif **4** in a range between $\lambda = 300$ and $\lambda = 420$ nm, showing two sets of curves for the reversible photo switching of the DMAB moiety for alternating irradiation with UV (solid lines) and visible light (vis., dotted lines). The absorption maximum lies at $\lambda = 345$ nm. **d** Analytical PAGE analysis of reversible switching 2′,6′-dimethylazobenzene functionalized **DxR**-L4 motif **4**. **e** Fluorescence emission spectra ($\lambda_{ex} = 480$ nm) of a **DxR** solution with increasing molar ratios of **4** in the range of 1–7 μM (0.1–0.7 equiv.) showing a reduction in fluorescence intensity of **DxR** with an increasing concentration of added motif **4**. **f** Comparison of fluorescence quenching of **DxR** with the DMAB moiety in *trans*- (black circles) and in *cis*- (white squares) conformation ($n = 3$, error bars: mean ± S.E.M.)

thereby introducing four lipid tails into each aptamer. The resulting lipid-functionalized aptamer trCLN3-L4 (**3**) was purified by reversed-phase HPLC (Supplementary Fig. 4) and confirmed by LCMS mass spectrometry (Supplementary Fig. 5). Polyacrylamide gel electrophoresis (PAGE) of lipidated and non-lipidated trCLN3 aptamers showed significant differences in the migration behavior, consistent with L4-modification (data not shown). Moreover, the L4-modified aptamers showed a strong tendency to self-aggregate in aqueous solution by forming spherical nanoconstructs above a critical micelle concentration (CMC) at room temperature. We evaluated the CMC of the trCLN3-L4 nanoconstructs using Förster resonance energy transfer (FRET; Supplementary Methods; Supplementary Fig. 6; Supplementary Table 1)[65,66] and fluorescence studies with pyrene-loaded trCLN3-L4 nanoconstructs (Supplementary Fig. 7; Supplementary Table 2)[67]. Both methods yielded CMC values in the range of 300–350 nM concentrations. The size and morphology of the nanoconstructs were further studied by atomic force microscopy (AFM; Fig. 1c) and electron microscopy (TEM; Supplementary Fig. 8). To obtain a statistical evaluation of the size distribution of nanoconstructs, the diameters of at least 50 nanoconstructs for each AFM image were compiled in histograms and fitted by Gaussian distributions (Fig. 1d). The trCLN3-L4 **3** nanoconstructs have an average diameter of $21.2 \pm 1.5$ nm (Fig. 1d), consistent with the size of 25 nm measured by TEM.

**Effect of lipid modifications on cMet binding and serum nuclease stability.** To test the influence of lipid tails on aptamer binding, a competitive filter-binding assay was carried out. Varying concentrations of unlabeled 5′-lipid-functionalized aptamer **3** and its two-point mutant variant trCLN3.mut-L4 (Supplementary Methods) competed with a constant amount of [32]P-labeled native trCLN3 aptamer in binding to cMet. Strong cMet binding was observed for trCLN3-L4 with an $IC_{50}$ value of 43 nM, compared to 56 nM obtained for the non-lipidated native aptamer trCLN3 (Supplementary Fig. 9). This result demonstrates that aptameric nanoconstructs retained their binding affinity to cMet as compared to the non-modified aptamer trCLN3. In contrast, the lipidated mutant aptamer trCLN3.mut-L4 containing two point mutations could not compete with the [32]P-trCLN3 for binding to cMet within the tested concentration range, indicating that the displacement of the non-lipidated [32]P-trCLN3 from its bound cMet-target by its lipidated counterpart trCLN3-L4 is specific.

Since an adequate serum half-life is a prerequisite for the successful in vivo application of these aptamers, the serum stabilities of aptamer trCLN3, its double point mutant non-binding variant trCLN3.mut, and their corresponding lipid-functionalized derivatives (trCLN3-L4 and trCLN3.mut–L4, respectively) were analyzed in 90% PBS-buffered fetal calf serum (FCS) (Fig. 2a) and in freshly prepared human blood serum

**Table 1 Atto-labeled motifs 3 and 4 mixed in different ratios to form HyApNc nanoconstructs**

| Exp. no. | Atto550-4 (μM) | Atto647N-3 (μM) | Volume (μL) | Equivalents Atto550-4 | $I_{669}$[a] (mean ± S.D.) | $I_{576}$[b] (mean ± S.D.) | $I_{669}/I_{576}$[c] |
|---|---|---|---|---|---|---|---|
| 1 | 0.0 | 5.0 | 20 | 0.0 | 652 ± 206 | 41 ± 7 | 15.90 |
| 2 | 1.0 | 5.0 | 20 | 0.2 | 2317 ± 657 | 416 ± 116 | 5.56 |
| 3 | 1.75 | 5.0 | 20 | 0.35 | 5673 ± 881 | 775 ± 169 | 7.32 |
| 4 | 2.5 | 5.0 | 20 | 0.5 | 9604 ± 1172 | 1218 ± 234 | 7.88 |
| 5 | 5.0 | 5.0 | 20 | 1.0 | 21,098 ± 402 | 3553 ± 434 | 5.93 |
| 6 | 7.5 | 5.0 | 20 | 1.5 | 28,225 ± 1164 | 6106 ± 378 | 4.62 |
| 7 | 10 | 5.0 | 20 | 2.0 | 34,010 ± 3593 | 9992 ± 153 | 3.40 |
| 8 | 15 | 5.0 | 20 | 3.0 | 35,242 ± 5951 | 27,766 ± 4606 | 1.26 |

[a]Fluorescence intensities at $\lambda = 669$ nm
[b]Fluorescence intensities at $\lambda = 576$ nm
[c]Estimated ratio ($I_{669}/I_{576}$) from the FRET experiments

(HBS, Fig. 2b) at 37 °C from 0 to 72 h (full uncropped PAGE included as Supplementary Figs. 10, 11). A comparison of degradation profiles between FCS and HBS (Fig. 2c) revealed fairly similar patterns of aptamer degradation for both serum samples. The non-lipidated variants of the aptamer samples degraded 1.5-fold faster in HBS compared to FCS. Under similar conditions the serum half-life ($t_{1/2}$) of trCLN3 was 8.7 h (90% PBS-buffered FCS) and 4.9 h (90% PBS-buffered HBS), respectively, compared to its lipid-functionalized derivative trCLN3-L4 showing no significant degradation even up to 72 h of incubation in both sera. To exclude the possibility that the differences in serum stability are due to the G-quadruplex present in both trCLN3-L4 and trCLN3, we also compared serum stabilities of trCLN3.mut-L4 and trCLN3.mut, both not capable of forming a G-quadruplex. The $t_{1/2}$ values of trCLN3.mut-L4 in FCS (~30.6 h) and in HBS (~36.8 h), respectively, was approximately 10-fold and 19-fold higher than that of the non-lipidated variant trCLN3. mut ($t_{1/2} = 2.8$ h in FCS; 1.9 h in HBS). These observations clearly indicate that the serum stability of the mutant aptamer is lower than that of trCLN3 native aptamer, and lipidation further protects the aptamer against enzymatic degradation thereby increasing the serum stability several fold.

**Design of a photoswitchable DxR-binding motif**. We next synthesized the thermodynamically stable lipid-modified DNA motif **4** (Fig. 1e) consisting of a preferred **DxR**-binding 37 nucleotide alternating GC sequence combined with four DMAB moieties and 4-lipid tails attached to the 5′-end (Fig. 3a, b). Motif **4** was designed to bind and release **DxR** reversibly by irradiating with UV or visible light (Supplementary Methods) and the integrity of the **DxR**-L4 motif **4** was confirmed by LC-MS (Supplementary Fig. 12). Reversible photoswitching of the four DMAB groups contained in motif **4** was investigated by UV/vis spectroscopy. The switching process is fully reversible and can be repeated for at least five irradiation cycles (Fig. 3c). This result is further supported by gel electrophoresis of the DMAB-modified GC-rich hairpin structure that showed a change in electrophoretic shift upon repeated irradiation with UV and visible light for 5 min each (Fig. 3d; Supplementary Fig. 13), consistent with significant structural changes between the hairpin and dehybridized motif.

The goal of intercalating and efficiently delivering multiple **DxR** molecules per motif **4** was investigated by binding studies between motif **4** and **DxR**. A fixed concentration (10 μM) of **DxR** was incubated with an increasing molar ratio of motif **4** (1−7 μM) and fluorescence quenching due to intercalation of **DxR** was used to examine the binding efficiency. Gradual decrease of the fluorescence intensity of **DxR** was observed upon binding to increasing amounts of motif **4** (Fig. 3e). We further tested the

difference in binding affinity of motif **4** for the *cis*- and *trans*-conformations of the DMAB groups. To do so, motif **4** was separately irradiated with visible light ($\lambda = 450$ nm) and UV light ($\lambda = 365$ nm) for 5 min each and mixed with a fixed concentration of **DxR** (10 μM), while the concentration of motif **4** was varied from 0.1−0.7 equivalents to that of the **DxR** concentration. The fluorescence curve of motif **4** with DMAB in *trans*-conformation ($\lambda = 450$ nm) showed a higher reduction in fluorescence intensity with an increasing molar equivalent of added motif **4** as compared to **4** in which the DMAB moieties were in *cis*-conformation. The difference in fluorescence intensity is about 30% higher in case of *trans*-DMAB than in *cis*-DMAB (Fig. 3f). This difference in fluorescence intensities further supports the conclusion, that the DMAB-modified motif **4** is destabilized by irradiation with UV-light thereby releasing **DxR**.

Next, we evaluated the percentage of **DxR** bound to motif **4**. A fixed amount of motif **4** (5 μM) intercalated with a 10-fold excess of **DxR** for 12 h followed by a purification step using spin filtration (Supplementary Methods). After each centrifugation step, a UV/vis spectrum of the flow through washing was recorded. A 20% reduction in **DxR** absorbance confirmed that approximately eight equivalents of **DxR** intercalate per motif **4**, and that two equivalents of excess **DxR** are removed through repeated washing (Supplementary Fig. 14).

We then quantified the **DxR** release from the loaded **DxR**-L4 motifs under photoirradiation by an HPLC assay, detecting the fluorescence of the remaining **DxR** bound to motif **4** after removing unbound excess **DxR** from the solution. Phenol/CHCl$_3$ (1:1) washing is known to remove unbound excess **DxR** in the presence of DNA duplexes without removing the intercalated **DxR**[68]. We then compared the amount of released **DxR** to that observed by self-diffusion of **DxR** into the buffer medium incubated at 37 °C over time (Supplementary Fig. 15a, b). After 5 min of UV irradiation ($\lambda = 365$ nm, 350 mW cm$^{-2}$), an approximately 3-fold drop in fluorescence emission was observed for the irradiated sample compared to the non-irradiated sample. Thus, UV irradiation triggered a rapid release of 63% of the encapsulated **DxR** (Supplementary Fig. 15a). In contrast, a non-irradiated sample incubated at 37 °C released only about 20% of the loaded **DxR** from motif **4** over 48 h of incubation, due to thermal self-diffusion (Supplementary Fig. 15b). To compare the UV-induced **DxR** release to thermally driven **DxR** diffusion at a fixed time interval, aliquots of sample incubated at 37 °C for 48 h were analyzed before and after irradiation with 365 nm UV light for 5 min. The release of **DxR** was monitored by measuring the fluorescence of irradiated vs. non-irradiated sample at 590 nm using a fluorescence detector attached to HPLC. **DxR**-loaded motif **4** incubated at 37 °C without UV exposure led to a release of 20% of the loaded **DxR** within 48 h of incubation by thermal self-diffusion. The same sample, however, released an additional

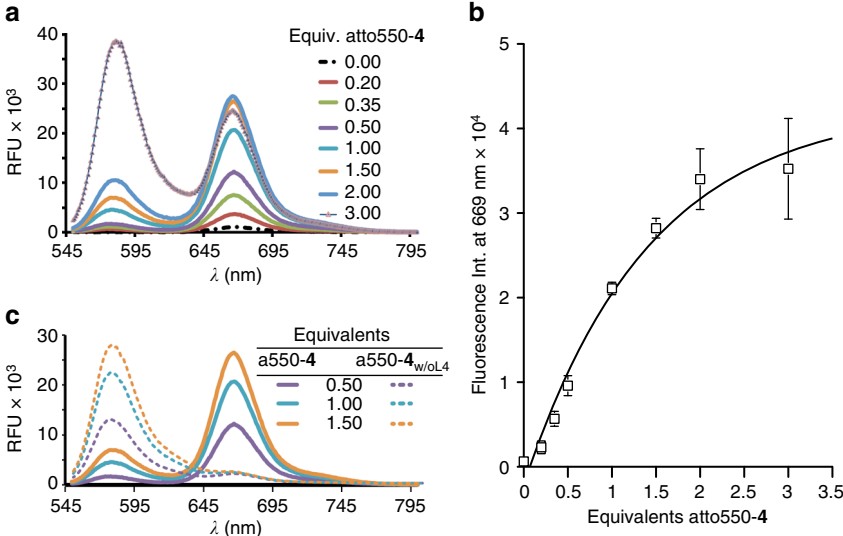

**Fig. 4** FRET study of the formation of functional hybrid-nanoconstruct HyApNc. **a** Fluorescence emission spectra ($\lambda_{ex} = 535$ nm) for FRET-assembled Atto647N-labeled trCLN3-L4 (Atto647N-**3**) and Atto550-labeled **DxR**-L4 motif (Atto550-**4**) HyApNc formation. Atto647N-**3** was kept constant at 5 µM with increasing equivalents of Atto550-**4**. **b** Maximum fluorescence intensities at $\lambda = 669$ nm ($I_{669}$) as a function of increasing concentration of Atto550-**4** showing an increase in energy transfer ($n = 3$, error bars mean ± S.D.). Saturation is reached between 2.0 and 2.5 equivalents of Atto550-**4**. **c** Comparison of the FRET signal ($\lambda_{ex} = 535$ nm) of HyApNc consisting of **4** (straight) and **4** without the lipid tail (a550-**4**$_{w/oL4}$; dashed)

50% of the loaded **DxR** immediately after UV irradiation (Supplementary Fig. 15b, black square). These results support the conclusion that the release of **DxR** from the motif **4** is stimulated by UV irradiation.

**Lipid-mediated self-assembly of motifs 3 and 4 forms HyApNc.** We next combined both lipid-modified motifs **3** and **4** to test their lipid-mediated self-assembly into heterogeneous HyApNc. By mixing free Atto-647N-trCLN3-L4 (Atto647N-**3**) with Atto550-labeled **DxR**-L4 motif (Atto550-**4**) in different ratios, hybrid nanoconstructs were formed and stabilized by the strong hydrophobic interaction of the lipid tails. The Atto-dye labels were attached at the 5′-end in immediate proximity to the lipid modifications to ensure that intermolecular FRET effects report the formation of micellar nanoconstructs. In the FRET experiment nanoconstructs self-assembled by mixing a fixed concentration of 5 µM Atto647N-**3** with Atto550-**4** in concentrations ranging between 1 and 15 µM (Table 1).

Fluorescence at $\lambda_{ex} = 535$ nm (Fig. 4a) showed that the nanoconstructs self-assembled with 0.2 equivalents of Atto550-**4** (Atto647N-**3**:Atto550-**4** = 5:1), yielding an intensity ratio $I_{669} \times I_{576}^{-1}$ of 5.56. In contrast, nanoconstructs self-assembled with 0.35 or 0.5 excess equivalents of Atto550-**4** showed an increasing $I_{669} \times I_{576}^{-1}$ value of 7.32 and 7.88, respectively, a significant enhancement of ~32% and ~41% relative to the Atto647N fluorescence. An increase in FRET observed with increasing concentrations of Atto550-**4** reached saturation between 2.0 and 2.5 equivalents (Fig. 4b). Nevertheless the $I_{669} \times I_{576}^{-1}$ value already reaches 5.93 at one equivalent of Atto-550-**4** (Atto647N-**3**:Atto550-**4** = 1:1). Therefore, we maintained this ratio in the subsequent cellular studies to achieve a proper balance between high target affinity (internalization efficiency) and **DxR** carrying efficiency (cytotoxicity).

In a control experiment, we employed the Atto550-labeled **DxR**-binding motif without lipid modification (a550-**4**$_{w/oL4}$). With this lipid-devoid motif, only diffusion-controlled encounters between Atto550 and Atto647N can occur, which should result in very low relative intensities. Indeed, with a 1:1 ratio of

motif **3** and Atto550-**4**$_{w/oL4}$ we observed an $I_{669} \times I_{576}^{-1}$ value of 0.09, indicating that no hybrid micellar nanoconstructs are forming (Fig. 4c). The FRET signal thus strictly depends on the ratio of the two functional domains and on the presence of the L4 modification. A comparison of FRET efficiency values (Supplementary Methods; Supplementary Fig. 16) suggested the 92% FRET efficiency for assembled HyApNc consisting of motifs Atto550-**4** and Atto647N-**3** as compared to 27% where both motifs **4** and **3** lack the lipid modifications. When a non-cMet-binding Atto647N-labeled mutant trCLN3-L4 motif (Atto647-mut-**3**) was used instead of Atto647N-**3**, the resulting mutated nanoconstruct HyApNc.mut yielded a FRET efficiency (97%) similar to HyApNc. Together, these data provide evidence that both motifs self-assemble to form hybrid heterogeneous nanoconstructs of spherical geometry when the lipid modifications are present. The FRET signal intensity is also a good measure of integrity of the nanoconstructs.

The resulting HyApNc consisting of **3** and **4** in a 1:1 ratio was further analyzed by AFM to compare its size and structural features with nanoconstructs resulting only from motif **3**. We observed that the hybrid micellar nanoconstruct retained its spherical shape similar to the homogenous nanoconstructs consisting of only motif **3** (Fig. 1c, f). However, their average diameter is 32.3 ± 2.1 nm—larger than the homogenous nanoconstructs made from trCLN3-L4 (motif **3**), which averaged 21.2 ± 1.5 nm (Fig. 1d, g). This increase in size of the heterogenous nanoconstructs as compared to the homogenous ones may result from differences in the physico-chemical properties of the two aptamers in **3** and **4**, or from structural differences, or both.

For efficient cell internalization and for successful delivery of the intercalated **DxR**, the integrity of the mixed micellar nanoconstructs HyApNc, i.e., both motifs **3** and **4**, must remain intact as a single nanoconstruct and circulate in complex biological media for a sufficiently long period of time. The stability of the micelles as well as their circulation time is known to be affected by the presence of serum proteins, which alter the micellar equilibrium leading to their dissociation to varying extents[69,70]. Therefore, we evaluated the integrity of HyApNc upon interaction with HBS, and in presence of bovine serum

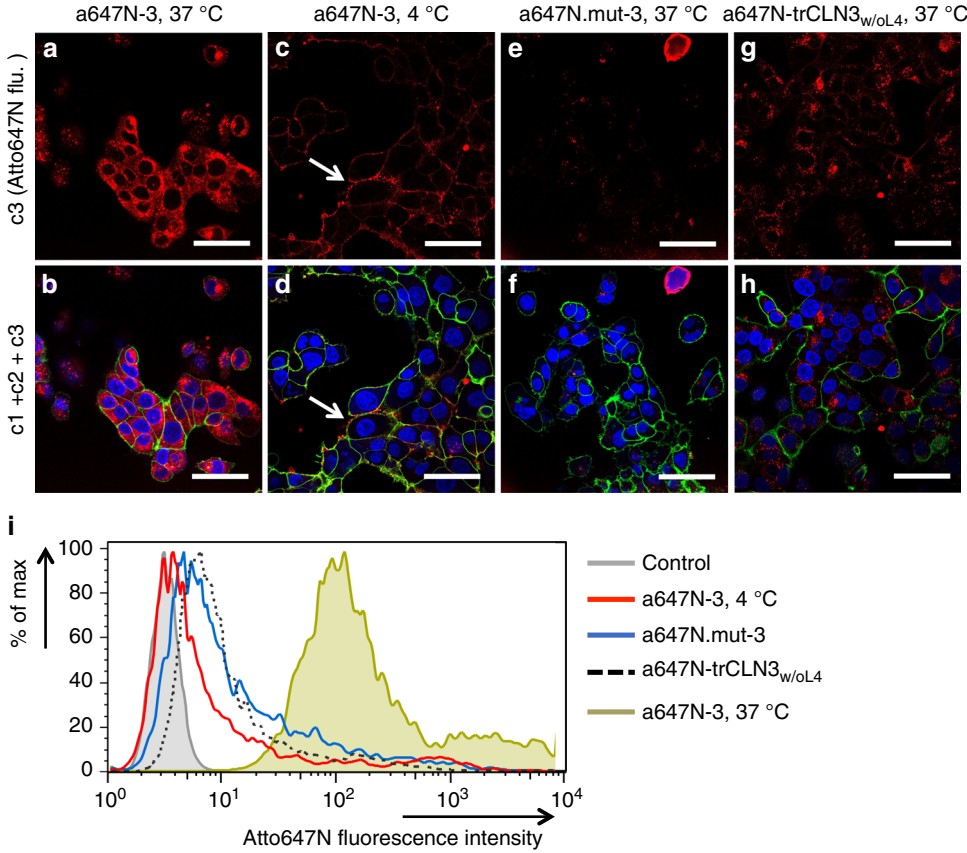

**Fig. 5** Confocal microscopy and flow cytometry analysis of Atto647N-trCLN3-L4 internalization. **a**, **b** Confocal images of NCI-H1838 cells incubated with Atto647N-labeled trCLN3-L4 (A647N-**3**) nanoconstructs at 37 °C (**a**: unmerged; red; c3 and **b**: overlay; c1 + c2 + c3). **c**, **d** NCI-H1838 cells incubated with Atto647N-**3** at 4 °C (**c**: unmerged; red; c3 and **d**: overlay; c1 + c2 + c3). Arrow: Alexa488-WGA membrane stain (green) shows colocalization with Atto647N-**3** (red). **e**, **f** NCI-H1838 cells treated with Atto647N.mut-**3** at 37 °C (**e**: unmerged; red; c3 and **f**: overlay; c1 + c2 + c3). **g**, **h** NCI-H1838 cells treated with Atto647N-trCLN3$_{w/oL4}$ (without lipid modification) at 37 °C (**g**: unmerged; red; c3 and **h**: overlay; c1 + c2 + c3). Cells were membrane stained with Alexa488-WGA (green; c2), nuclei were stained with Hoechst 33342 (blue; c1) and analyzed for Atto647N-**3** uptake (red; c3). Scale bars: **a**–**h** 50 µm. **i** FACS histograms for cells treated with Atto647N-**3** at 37 °C (green shadow) showed a significant shift in Atto647N fluorescence intensity compared to cells treated with Atto647N-**3** at 4 °C (red solid line), suggesting an endocytotic internalization pathway. A minimal shift in Atto647 fluorescence intensity was observed for cells treated with either a scrambled aptamer Atto647N.mut-**3** (blue solid line) or with Atto647N-trCLN3$_{w/oL4}$ (black dotted line) at 37 °C compared to untreated cells (gray shadow), indicating a marginal internalization presumably due to non-specific binding or lack of lipidation

albumin (BSA) at 37 °C over time (Supplementary Methods; Supplementary Fig. 17a, b, c). We assessed the integrity of the micellar nanoconstruct HyApNc by using the previously assembled FRET pair (see Fig. 4) attached to the 5′-ends of both motifs **3** (Atto647N-**3**) and **4** (Atto550-**4**). The intermolecular FRET effect was monitored (Supplementary Fig. 17a, b) and an increase in the fluorescence intensity at 576 nm and a decrease at 669 nm was observed over time. This result indicates that the micellar nanoconstructs disintegrate gradually in the presence of BSA or serum proteins contained in HBS. The FRET ratio = $I_{669} \times (I_{669} + I_{576})^{-1}$ was calculated and plotted as a function of time (Supplementary Fig. 17c). The HyApNc nanoconstructs exhibited a half-life ($t_{1/2}$) of 14 h in 95% HBS and of 18 h in 1 mM BSA solution. The time-resolved emission data indicate that the rate of micellar nanoconstruct disintegration in either BSA or HBS was not significantly different. The $t_{1/2}$ indicates an adequate stability of the micelles in blood serum with slow disintegration under our in vitro experimental conditions. It is possible that the $t_{1/2}$ of HyApNc will be further reduced in the blood stream in vivo. However, if necessary for in vivo applications the half-life of HyApNc could be further increased by elongating the lipid chains and/or by using unsaturated lipids and crosslinking them at the core of the nanostructures.

**Uptake of aptameric nanoconstructs by cMet-expressing cells**. After confirming the successful fomation of the aptameric nanoconstructs, the cell targeting ability and internalization efficacy of aptamer trCLN3-L4 (**3**) mediated by cMet recognition was investigated by using both confocal microscopy and flow cytometry analysis. Cell uptake experiments were performed with the NCI-H1838 lung cancer cell line that expresses cMet[64]. NCI-H1838 cells incubated with different concentrations of the Atto647N-**3** (10 and 1 µM, respectively) at 37 °C for 90 min, showed a strong and comparable intracellular red fluorescence at both concentrations above the CMC value (Fig. 5a, b and Supplementary Fig. 18a, b for 10 µM; Supplementary Fig. 18c, d for 1 µM). At 1 µM of Atto647N-**3**, a punctuated pattern of internalized nanostructures was observed in the cytoplasm, suggesting that they may localize in endosomes (Supplementary Fig. 18c, d). Indeed, the same experiment performed at 4 °C showed only a weak membrane-localized fluorescence (Fig. 5c, d) with markedly reduced Atto647N-fluorescence in the H1838 cells, consistent with inhibition of endocytosis at low temperature[71]. When the Atto647N-**3** concentration was reduced to 0.2 µM, which is below the CMC, a significantly weaker fluorescence signal was observed, as expected (Supplementary Fig. 18e, f).

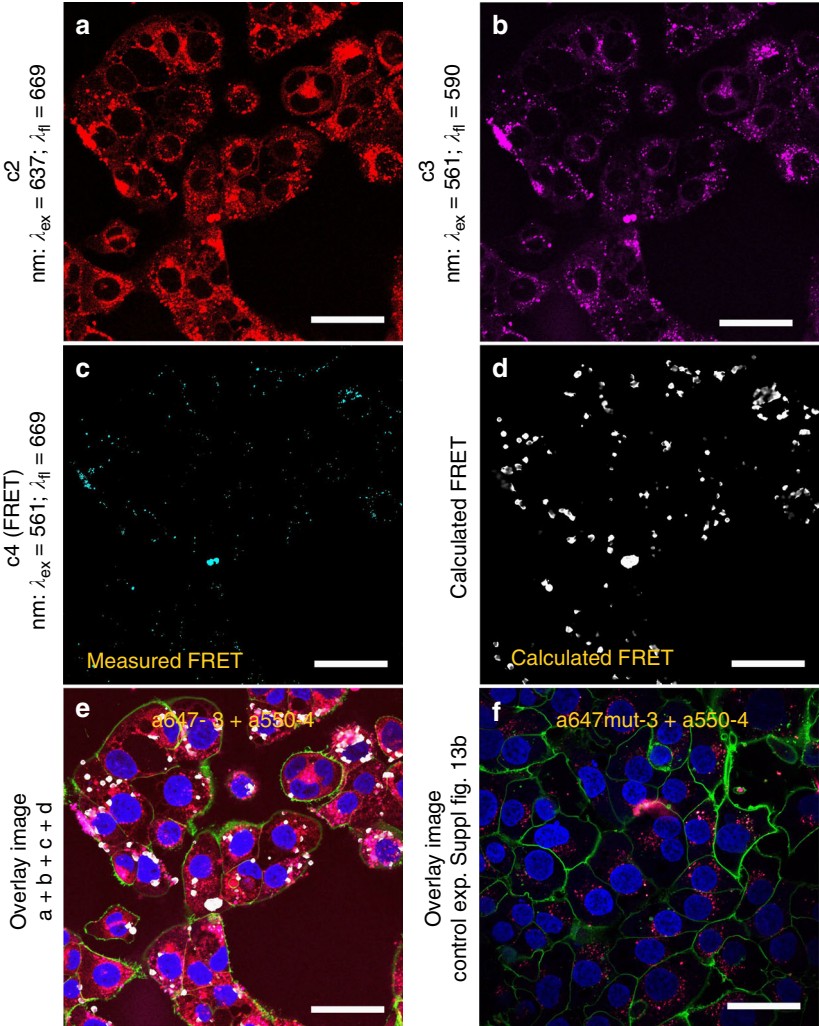

**Fig. 6** Cellular uptake of dual-labeled HyApNc consisting of A550-**4** and A647N-**3** motifs. **a–c** Confocal fluorescence images of H1838 cells treated with the HyApNc consisting of Atto550-**DxR**-L4 motif (A550-**4**) and Atto647N-trCLN3-L4 (A647N-**3**) motifs in 1:1 ratio. Both A647N-**3** (**a**: c2: red) and A550-**4** (**b**: c3: magenta) fluorescence were observed from the cytosol including a FRET-mediated Atto647N signal (**c**: c4: cyan). **d** Calculated FRET signal from reconstructed FRET images (**d**: white) indicate the intracellular integrity of the functional nanoconstruct HyApNc. **e, f** Overlay images of cells incubated with HyApNc (**e**: A647N-**3** + A550-**4**), and HyApNc.mut (**f**: A647N.mut-**3** + A550-**4**) as a negative control with Atto647N-labeled mutant trCLN3.mut-L4 motif. The complete overlay sets for **e** and **f** are shown in Supplementary Fig. 19. Aptamer constructs were incubated at 37 °C for 2 h, followed by membrane staining with Alexa488-WGA (green), and nuclei staining with Hoechst 33342 (blue). Scale bar: **a–f** 50 μm

H1838 cells incubated with 5′-Atto647N-labeled double mutant of **3** (Atto647N-mut **3**) that does not bind to cMet exhibited marginal cellular staining (Fig. 5e, f), consistent with lack of internalization. Finally, the non-lipidated version of Atto647N-trCLN3$_{w/oL4}$ also showed low cellular staining (Fig. 5g, h), suggesting that lipidation of the cMet-binding aptamer is required for efficient uptake. This result may indicate that protein target binding in solution could differ from targeting the protein at the cell surface. Moreover, lipidation of aptamers potentially improves their ability to target proteins expressed on cell surfaces by self-organizing multiple aptamers in a single nanostructure, although the generality of this notion remains to be demonstrated with other aptamer/target systems.

These findings were further confirmed through flow cytometric studies (Fig. 5i; Supplementary Methods). There was a noticeable change in the fluorescence signal observed for cells treated with free Atto647N-trCLN3$_{w/oL4}$ (Fig. 5i, black dotted) compared to the auto-fluorescence profile of untreated cells (Fig. 5i, gray area), indicating low internalization. Compared to non-lipidated Atto647N-trCLN3$_{w/oL4}$, cells treated with Atto647N-**3** at 37 °C

(Fig. 5i, green area) showed significantly higher shift in fluorescence intensity. A minimal shift in fluorescence intensity was also observed for cells treated with either Atto647N-mut **3** (Fig. 5i, blue curve) or Atto647N-**3** at 4 °C (Fig. 5i, red curve) over untreated cells (gray areas), indicating either a low non-specific binding or only a membrane localized binding without internalization at low temperature. Taken together, these results show that important determinants for efficient uptake into H1838 cells are the ability to bind extracellular cMet by the aptamer moieties, the ability to form nanoconstructs due to lipidation, and that the uptake is temperature-dependent, supporting an endocytotic mechanism.

We next performed cellular uptake studies of a dual-labeled hybrid-nanoconstruct (HyApNc) containing a mixture of Atto550-labeled **4** and Atto647N-labeled **3** motifs in a 1:1 ratio. Both fluorescent probes constitute a suitable FRET pair entrapped within the lipid core that can be employed to validate whether the functional nanoconstructs enter and target H1838 cells. The confocal images showed not only the cellular staining for both dyes (Fig. 6a; c2: red, Fig. 6b; c3: magenta), but also a FRET signal

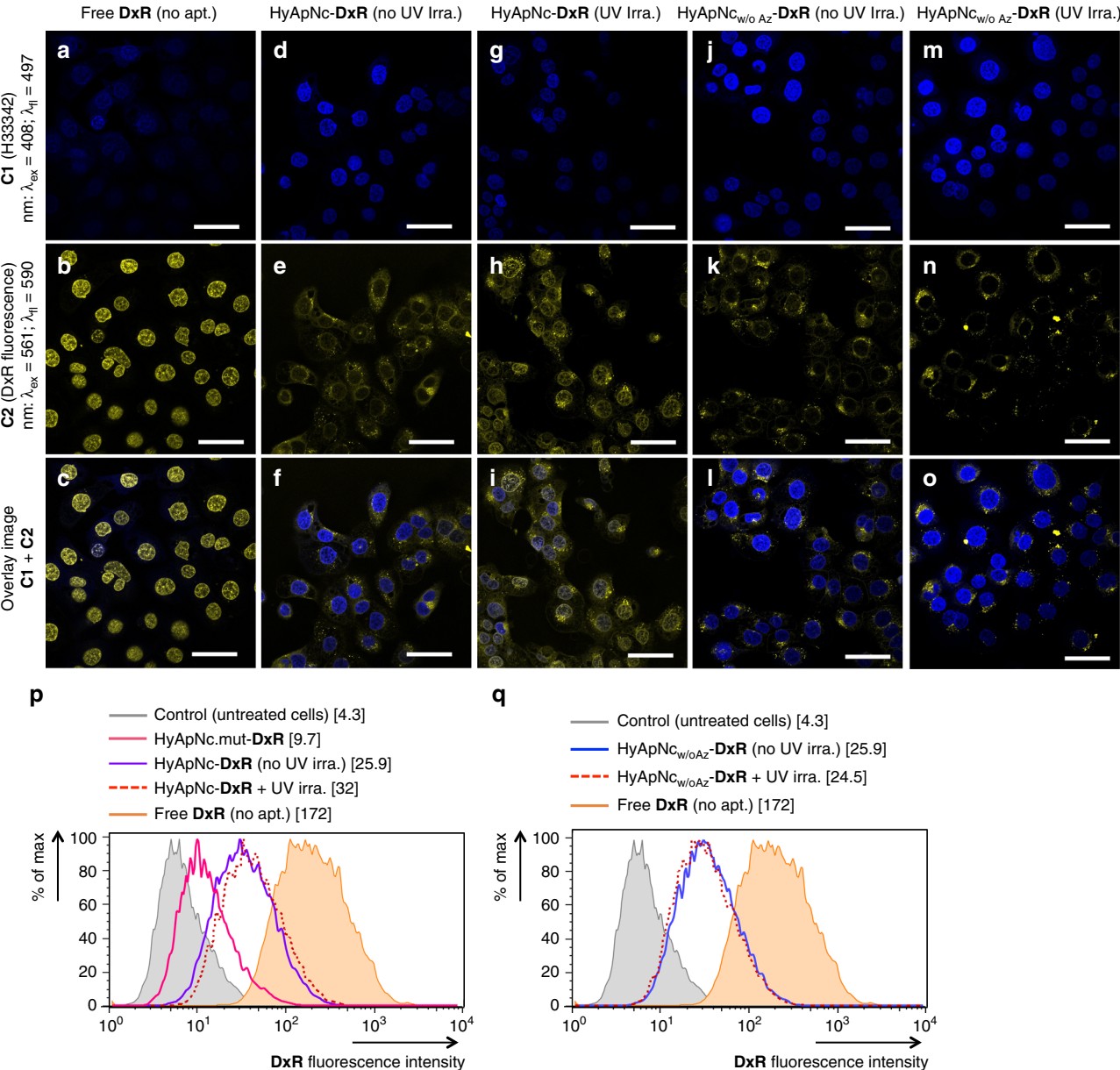

**Fig. 7** Confocal microscopy and flow cytometry analysis of the HyApNc-mediated doxorubicin uptake. **a–c** Confocal image of intracellular distribution of **DxR** released (yellow signal) from the **DxR**-loaded HyApNc nanoconstructs in the H1838 cells incubated with free **DxR** (**a**: nuclei staining with Hoechst 33342; blue; c1, **b**: **DxR** staining; yellow; c2, **c**: overlay; c1 + c2). **d–f** Cells incubated with HyApNc-**DxR** not exposed to UV irradiation (**d**: Hoechst 33342; blue; c1, **e**: **DxR** staining; yellow; c2, **f**: overlay; c1 + c2). **g–i** Cells incubated with HyApNc-**DxR** exposed to UV light, $\lambda = 365$ nm, 350 mW cm$^{-2}$ (**g**: Hoechst 33342; blue; c1, **h**: **DxR** staining; yellow; c2, **i**: overlay; c1 + c2). **j–l** Cells treated with HyApNc$_{w/oAz}$-**DxR** without UV irradiation (**j**: Hoechst 33342; blue; c1, **k**: **DxR** staining; yellow; c2, **l**: overlay; c1 + c2). **m–o** Cells treated with HyApNc$_{w/oAz}$-**DxR** exposed to UV light ($\lambda = 365$ nm, 350 mW cm$^{-2}$) (**m**: Hoechst 33342; blue; c1, **n**: **DxR** staining; yellow; c2, **o**: overlay; c1 + c2). Blue (c1) and yellow (c2) signals show the fluorescence of Hoechst 33342 and **DxR** staining, respectively. The overlay (c1 + c2) shows colocalization of Hoechst 33342 and **DxR**. An increase in nuclear accumulation of **DxR** upon light triggering was observed only for the photoactivated nanoconstruct. Scale bar: **a–o** 50 μm. **p** Flow cytometry histogram showing quantitative comparison of **DxR** accumulation in H1838 cells after incubation with free **DxR** (orange shadow), mutant non-targeted nanoconstructs HyApNc.mut-**DxR** (magenta solid), targeted nanoconstructs HyApNc-**DxR** without UV (purple solid), or with UV irradiation (red dotted). **q** Flow cytometry histogram showing **DxR** accumulation in H1838 cells after incubation with HyApNc$_{w/oAz}$-**DxR** without UV (blue solid) or with UV irradiation (red dotted) at 37 °C for 2 h. The concentration of **DxR** either in free form or its equivalent in complex form in the cell culture was fixed at 8 μM. Untreated cells are shown in shadow (gray). The numbers in bracket of the legends are the geometric mean of the corresponding peaks

(Fig. 6c; c4: cyan) was observed. During confocal imaging all settings were kept constant (for details, see Methods). To evaluate the occurrence of FRET, we analyzed the images using a method that was previously reported[72], where the PixFRET plugin of the image processing software ImageJ was used for FRET quantification. Briefly, the bleed-through of the acceptor and donor

channels was determined and finally the calculated FRET images were reconstructed (Fig. 6d; white: calculated FRET). The calculated FRET images suggest donor and acceptor dyes are in correct geometry, supporting the integrity of the nanoconstructs. High FRET efficiencies were only observed when the designated constructs were able to enter the cells (Fig. 6a–c). Figure 6e shows

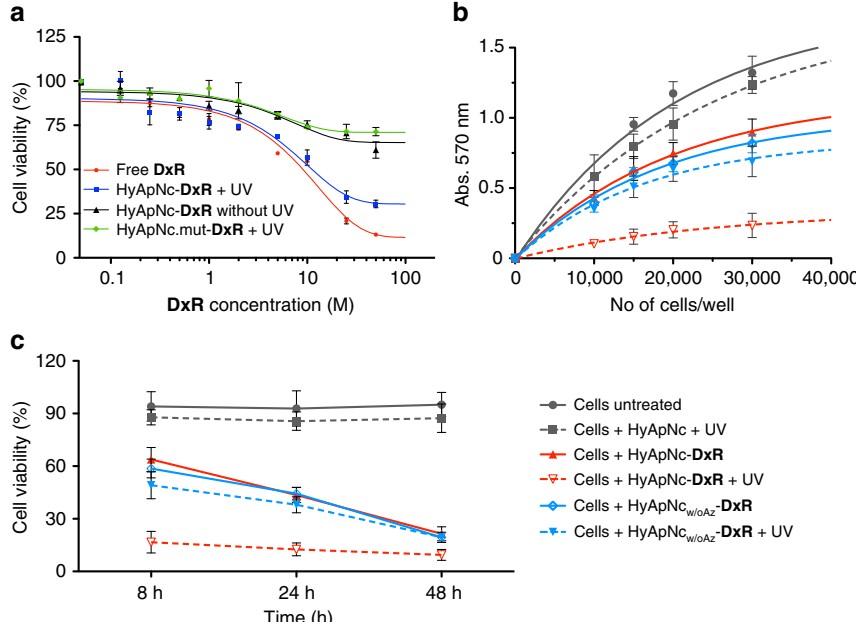

**Fig. 8** Cell viability assays of **DxR**-loaded HyApNc nanoconstructs in NCI-H1838 cells. **a** Cytotoxicities of HyApNc–**DxR** and HyApNc.mut–**DxR** complexes in combination with UV irradiation at the indicated **DxR** concentrations (0.125–50 μM ranges). As a control, viabilities of the cells treated with free **DxR** alone and HyApNc–**DxR** complex without UV irradiation were compared (n = 2, error bars: mean ± S.D.). **b** 8 h post incubation MTT assays where an increasing number of H1838 cells treated with unloaded HyApNc (gray), photoactive HyApNc-**DxR** (red), and photo-inactive HyApNc$_{w/oAz}$-**DxR** (blue) with and without subsequent UV irradiation (dashed vs. solid lines). As control, cell viabilities of the H1838 cells treated with RPMI medium with 10% FCS and not exposed to UV irradiation (gray solid) were measured at 570 nm (n = 2, error bars: mean ± S.D.). **c** Time-dependent cytotoxicities of photoactive HyApNc-**DxR** (red) against photo-inactive HyApNc$_{w/oAz}$-**DxR** (blue) with and without UV irradiation (dashed vs. solid lines) where the cells were treated with the **DxR** complex for various incubation times of 8, 24, and 48 h, respectively, before being subjected to the MTT assay (n = 2, error bars: mean ± S. D.)

the overlay of the images shown in Fig. 6a–d (including nuclear and membrane staining). In contrast, mutated nanoconstructs (HyApNc.mut) containing the non-cMet-binding Atto647N-labeled trCLN3.mut-L4 motif and Atto550-labeled motif **4** resulted in poor FRET efficiencies (Fig. 6f, Supplementary Fig. 19f–j), similar to background signals, indicating that the process of internalization is target-specific rather than occurring randomly.

**Photo-triggered release of DxR from HyApNc-DxR.** After successfully targeting the H1838 cells with HyApNc, we further investigated the selective transport of **DxR** into the cells, followed by its light triggered release from the HyApNc. The **DxR**-loaded HyApNc (HyApNc–**DxR** complex) was prepared by mixing motif **3** and **4** (1:1 ratio) with 10-fold excess of **DxR** followed by a purification step using spin filtration (details are given in Methods). To ensure minimum cell mortality upon UV irradiation, H1838 cells were irradiated at t = 0, 5, 10, 15, and 30 min, respectively, at an intensity of 350 mW cm$^{-2}$. Cell viability as a function of time-dependent response to UV treatment was measured by an MTT assay 24 h after irradiation. A maximum survival rate comparable to the non-irradiated control (t = 0 min) was observed at an irradiation time t ≤ 5 min (Supplementary Fig. 20a, b).

To verify the HyApNc-mediated selective transport of **DxR** to target cells and its light-triggered release from motif **4**, we monitored the fluorescence signal of **DxR** within and outside of the cell nuclei that were treated with either free **DxR** (as control) or with HyApNc-**DxR** (details of **DxR** loading: see Methods), while keeping the **DxR** concentrations in the bound and the unbound form fixed at 40 μM (5 μM HyApNc carrying eight

equivalents of **DxR**). The release of **DxR** from HyApNc was investigated by confocal microscopy with and without subsequent irradiation at 365 nm. Confocal images of the H1838 cells at 37 °C after 2 h of incubation showed a decrease in the **DxR** fluorescence signal in the cell nuclei in the following order: free **DxR**, HyApNc–**DxR** complex with and without UV irradiation (λ = 365 nm, 350 mW cm$^{-2}$) (Fig. 7a–i). Strong **DxR** fluorescence was observed in cell nuclei after treatment with free **DxR**, indicating that free **DxR** readily diffuses through the plasma membrane and accumulates almost exclusively in the nuclear region (Fig. 7a–c). However, the HyApNc–**DxR** complex without UV irradiation led to a considerably weaker **DxR**-fluorescence in the nucleus and a noticeable fluorescence within the endoplasm confirming that most of the **DxR** is predominantly localized outside the nucleus bound to the HyApNc (Fig. 7d–f). In contrast, when the HyApNc–**DxR** complex is exposed to irradiation (λ = 365 nm, 350 mW cm$^{-2}$) a discernible increase in both nuclear and extranuclear fluorescence was detected (Fig. 7g–i). When control experiments were performed with a construct lacking DMAB (HyApNc$_{w/oAz}$-**DxR**), near-identical **DxR** fluorescence signals are predominantly observed in the cytosol of the cells with and without UV exposure (Fig. 7j–o). No visible increase in the **DxR** fluorescence signal was observed in either the nuclei or in the cytosol when the cells treated with HyApNc$_{w/oAz}$-**DxR** were irradiated (Fig. 7m–o) compared to non-irradiated cells (Fig. 7j–l).

To further validate these results, the HyApNc-mediated **DxR** internalization with or without DMAB was evaluated by flow cytometry. As a control, the **DxR** uptake of the non-targeted mutated nanoconstruct HyApNc.mut-**DxR** was compared to that of the targeted nanoconstructs HyApNc-**DxR** (Supplementary Methods). To accomplish this, H1838 cells were incubated with

free **DxR**, HyApNc.mut-**DxR**, HyApNc-**DxR**, or targeted nano-constructs without DMAB (HyApNc$_{w/oAz}$-**DxR**) at fixed **DxR** concentrations of 8 μM either in its free form or in its complex form with the carrier (1 μM of nanocarrier, each containing eight equivalents of **DxR**). Treatment of cells with free **DxR** (orange areas) induces a 5-fold increase in mean cellular fluorescence intensity as compared to cells incubated with an equivalent dose of either HyApNc-**DxR** (Fig. 7p, purple) or HyApNc$_{w/oAz}$-**DxR** (Fig. 7q, blue). Instead, irradiation of cells treated with HyApNc-**DxR** (Fig. 7p, red dotted) induces only about a 1.3-fold shift in the fluorescence intensity compared to the non-irradiated cells (Fig. 7p, purple solid). This small shift in the fluorescence intensity might be due to the limitations of the flow cytometer to discriminate between the nuclear and the extranuclear fluorescence signal. In contrast, cells incubated with HyApNc$_{w/oAz}$-**DxR** showed a −1.05-fold shift in fluorescence intensity, and the FACS profile of the irradiated sample (Fig. 7q, red dotted) was comparable to the non-irradiated samples (Fig. 7q, blue solid). Moreover, cells incubated with HyApNc-**DxR** exhibited a 2.8-fold increase in the mean fluorescence signal compared to cells treated with HyApNc.mut-**DxR** containing the same amount of **DxR** in either case (Fig. 7p, purple vs. magenta). This result clearly showed that non-targeted nanoconstructs HyApNc.mut-**DxR** exhibited significantly lower efficacy in **DxR** delivery, consistent with their lower level of cellular uptake compared to HyApNc-**DxR** observed in Fig. 6. Overall this result indicates that after UV irradiation, most of the intercalated **DxR** was released from HyApNc having DMAB units and subsequently transferred into the nuclei and colocalized with the Hoechst dye.

**In vitro cytotoxicity of HyApNc-DxR against NCI-H1838 cells**. Having verified that the **DxR** can be selectively transported into target cells, we evaluated the cytotoxicity of the free **DxR**, the HyApNc-**DxR**, and the non-targeting HyApNc.mut-**DxR** nano-constructs with and without UV irradiation in H1838 cells by an MTT assay (for details, see Methods) in a dose-dependent way between 0.125 and 50 μM (Fig. 8a). There was a clear dependence of the H1838 cell viability on the concentration of **DxR** (Fig. 8a). An IC$_{50}$ of 11 μM (6.5 μg mL$^{-1}$) was determined for HyApNc-**DxR** irradiated with UV light (Fig. 8a, blue), and a similar level of cytotoxicity (IC$_{50}$ = 8 μM [4.7 μg mL$^{-1}$]) was observed for free **DxR** (Fig. 8a, red). However, no significant cytotoxicity was measured when cells were either treated with HyApNc-**DxR** without UV (Fig. 8a, black) or with HyApNc.mut-**DxR** (Fig. 8a, green). Cells incubated with non-targeting HyApNc.mut-**DxR** with subsequent UV irradiation under the same conditions (Fig. 8a, green) exhibited about a 38% increase in cell survival compared to cells treated with HyApNc-**DxR** at 8 μM loaded **DxR** concentrations (green vs. blue), consistent with their lower level of cellular uptake compared to HyApNc-**DxR** observed in Fig. 6. This result suggests that the cMet-expressing H1838 cells effectively internalized HyApNc-**DxR** due to receptor-mediated endocytosis, while non-targeted nanoconstructs exhibited significantly lower efficacy.

As an additional control, we conducted a time-dependent cytotoxicity assay to determine whether **DxR** release would occur solely through self-diffusion after endocytosis (i.e., no UV radiation). To accomplish this, we used the DMAB lacking construct (HyApNc$_{w/oAz}$-**DxR**) at different incubation times. H1838 cells were treated with unloaded HyApNc, HyApNc-**DxR**, and HyApNc$_{w/oAz}$-**DxR** for 2 h at 37 °C at 8 μM **DxR** dosage. After 2 h post treatment, the cells were washed, the RPMI medium replaced with fresh medium, and some of them (Fig. 8b, dotted) were exposed to UV light for 5 min (λ = 365 nm; 350 mW cm$^{-2}$), while those that were not irradiated were used as controls

(Fig. 8b, solid). Afterward cells were further allowed to incubate at 37 °C for 8, 24, and 48 h, respectively, before being subjected to the MTT assay. Cells treated with only RPMI medium and not exposed to UV irradiation (Fig. 8b, gray solid) served as the primary control.

Cells treated with HyApNc alone in combination with UV irradiation exhibited similar survival rates as non-irradiated cells treated with only RPMI medium (Fig. 8b, gray dotted vs. solid), indicating that neither the nanoconstruct without **DxR** nor brief UV exposure contribute significantly to cell death. In contrast, the combination of HyApNc-**DxR** with UV irradiation induced an approximately 2.8-fold decrease of cell viability compared to the treatment with HyApNc-**DxR** alone (17 vs. 64%) 8 h post treatment (Fig. 8b, red dotted vs. red solid). When cells were treated with the photo-deactivated construct HyApNc$_{w/oAz}$-**DxR** in combination with UV light a 0.2-fold decrease of cell viability compared to non-irradiated HyApNc$_{w/oAz}$-**DxR** (49 vs. 59%) was measured (Fig. 8b, blue dotted vs. blue solid). This result indicates that the lower cell mortality is related to inefficient release of **DxR** from the nanoconstrct without DMAB phtoswitches. We further evaluated cell viability for the incubation times of 24 and 48 h under similar conditions as for the 8 h incubation. Cells incubated with HyApNc-**DxR** without UV irradiation (Fig. 8c, red solid) showed a gradual decrease in viability from 64% (8 h) to 43% (24 h) to 21% (48 h). Cells incubated with HyApNc$_{w/oAz}$-**DxR** under the same conditions (Fig. 8c, blue solid) decreased from 59% (8 h) to 44% (24 h) to 19% (48 h). When UV irradiation was applied to the HyApNc$_{w/oAz}$-**DxR**-treated cells (Fig. 8c, blue dotted), cell viability was similar. Thus, a clear differentiation between 5 min UV irradiation of HyApNc-**DxR** and all other conditions was only seen for the 8 and 24 h incubation times, whereas at 48 h incubation cells were killed equally efficient under all conditions that contained **DxR**. At 48 h, a sufficient amount of intercalated **DxR** might have diffused from the control-nanoconstructs or the non-UV-irradiated ones spontaneously, and induce cell killing equally efficiently. For UV-irradiated HyApNc-**DxR**, a ~80% cell mortality is already achieved within a significantly shorter time span of 8 h (Fig. 8c, red dotted).

## Discussion

By exploiting aptamer-mediated selective cell targeting, photo-induced structure switching, and lipid-mediated self-assembly, we have developed a hybrid aptamer-nanoconstruct as a molecular carrier system that allows selective transport of intercalated cytotoxic drugs to target cells and release of the payload under light irradiation. This design offers the possibility to self-assemble multiple functional domains at once into a single nanoconstruct, in our case the targeting ability of an anti-cMet aptamer, and an intercalated drug-carrying motif, compared to the limited possibility of introducing multiple functionalities into a single modified aptamer system through inherent synthetic efforts. Fluorescence studies with pyrene loading showed that the self-aggregated nanoconstructs were stabilized in aqueous solution through hydrophobic interaction of the lipids. The mixed nature of the nanoconstructs and their size was confirmed by FRET studies and AFM measurements. Indeed, such self-assembled structures even offer an unprecedented degree of control over the ratio of different functional domains based on the therapeutic requirements. Moreover, integrating multiple GC-rich hairpin-duplex motifs affords several folds of loading of **DxR** into a single nanoscaffold, thereby enhancing the payload capacity in comparison to a monomeric aptamer.

Confocal imaging and cell viability assays further demonstrated a highly efficient cell uptake of the designed HyApNc into H1838 cells and an improved effect on tumor cell targeting by releasing

**DxR** inside the cell by a light trigger. We do not mean to suggest that UV light triggering will be compatible with therapy under standard clinical practice, as the penetration depth of light is only a few millimeters. While this skin depth may be sufficient for some melanoma, a better choice would be azobenzene photo-switches that isomerize with red light that has significantly higher skin penetration depth[73]. Alternatively, fiber optic endoscopy might direct UV light to potential tumor sites deeper inside the body.

An important outcome of our study is that lipidation of the aptamer motifs leads to a stable nanoconstruct with high resistance against nucleases accompanied by a drastically improved cell uptake compared to the unmodified aptamer. Despite using UV light as a trigger for unloading **DxR**, the induced cell death was successful without killing the cell itself by irradiation. As such, the successful implementation of DMAB into a photo-responsive DNA motif to control the drug release in cells using UV light represents a significant step toward aptamer-based targeted therapeutics. On the other hand, the possible risks associated with UV light such as cellular damage and stability of biological systems can be avoided by using low-intensity irradiation for a short period of time as indicated by our experiments. This concept might be further limited when applying it in vivo where the significant systemic dilution of the nanoconstructs could disintegrate the micelles more rapidly below the CMC. A possible solution to this issue might be attaching longer lipid tails to the ODN motifs, or using unsaturated lipids and cross-linking them inside the lipid core.

Ultimately, our work addresses fundamental obstacles related to aptamer-mediated tumor targeting while designing a multi-functional nanoconstruct with improved nuclease stability, high target-binding affinity, and increased tumor uptake, essential prerequisites for next-generation aptamer-based targeted therapeutics. Taken together, all these combined features make this platform widely applicable for the simultaneous delivery of a variety of different regulatory molecules, such as AntagomiRs, small interfering RNAs, microRNAs, drugs, and other molecules with high specificity and efficiency to specifically block functions of disease-relevant biomolecules.

## Methods

**Materials**. All chemicals including **DxR** were purchased from Sigma-Aldrich unless otherwise specified and were used as received. cMet-Fc, which represents the ectodomain of cMet fused to the Fc domain of human IgG1, was purchased from R&D Systems. Wheat Germ Agglutinin, Alexa Fluor® 488 Conjugate, and Hoechst 33342 were purchased from Life Technologies (Grand Island, NY, USA). γ-$^{32}$P-labeled ATP (250 µCi) was purchased from PerkinElmer Health Science B. V., The Netherlands. T4 Polynucleotide kinase and 1 × polynucleotide buffer were obtained from New England Biolabs, Frankfurt a. M., Germany. Binding buffer used for the aptamer competition-binding assay was prepared by adding E. coli tRNA (Roche AG, Mannheim, Germany), BSA (Thermo Fischer Scientific) into Dulbeccos PBS (Gibco, Life Technologies).

All solvents, reagents, and building blocks for oligonucleotide synthesis were obtained from Proligo, Hamburg, Germany. The anti-cMet aptamer motif (trCLN3) and its lipid derivatives (trCLN3-L4 and trCLN3.mut-L4) were synthesized according to the phosphoramidite protocol using an ABI 3400 synthesizer (Applied Biosystems). **DxR**-carrying **DxR**-L4 modified with DMAB and C$_{12}$-lipid tails as well as the fluorescent-labeled (Atto647N-, Atto550-, and 6FAM) trCLN3-L4 and **DxR**-L4 motifs were purchased in HPLC purified form from Ella Biotech GmbH, Munich, Germany.

**Cell culture and confocal microscopy**. The human NSCLC cell line H1838 was obtained from the American Type Culture Collection (ATCC). Cell cultures were tested for mycoplasma contamination by using the PCR-based Venor®GeM Mycoplasma detection kit. Cells were grown in T-75 cm$^2$ flasks using RPMI 1640 (Invitrogen) supplemented with 10% FCS in a humidified atmosphere at 37 °C and 5% CO$_2$. Cell lines were subcultured twice a week at a ratio of 1:4 depending on the confluence and cell density was determined with a hemocytometer before each experiment. Cells were detached using 1 mL Trypsin-EDTA solution (Sigma-Aldrich) followed by neutralization with 25 mL of RPMI medium and the cells were collected by centrifugation for 5 min at 400 rpm.

In vitro cell imaging of the cell internalization studies were performed using fluorescence microscopy. Prior to each experiment one 70–80% confluent flask was trypsinized and suspended with 10 mL of cell medium. 10 µL of the cell solution was pipetted onto a haemocytometer and the cells were counted. Twenty-four hours prior to the internalization experiments approximately 10,000 NSCLC cells were seeded in 96-well glass bottom multiwell cell culture plates (MatTek® Corporation). The plates were then incubated for 24 h at 37 °C in 5% CO$_2$ atmosphere. After 24 h of incubation the cells were first washed with 1× PBS buffer and incubated with various labeled aptameric nanoconstructs (trCLN3-L4, trCLN3.mut-L4, HyApNc-**DxR**, HyApNc.mut-**DxR**, or free **DxR**) in 100 µL of RPMI 1640 with 10% FCS medium containing 1 mM MgCl$_2$ at 37 and 4 °C separately for 2 h. The final concentrations of the labeled micelles were fixed at 10 µM. Afterward, cells were washed with fresh medium and Dulbeccos 1× PBS followed by 10 min fixation with 200 µL of a 3.7% (w/v) paraformaldehyde solution in Dulbeccos 1× PBS. Fixed cells were washed with fresh medium and Dulbeccos 1× PBS followed by staining with 200 µL of nuclear and plasma membrane staining reagent [60 µL (1 mg mL$^{-1}$) of Alexa Fluor 488-WGA and 20 µL of Hoechst 33342 (1 mM) in 4.0 mL in 1× PBS buffer] and incubated for 10 min at 37 °C. After 10 min, the labeling solutions were removed and the stained cells were washed with 1× PBS (2 × 200 µL) followed by addition of 200 µL of 1× PBS buffer. Finally, the 96-well plate was mounted with a multi-well plate holder and the confocal imaging of the fixed cells was performed by using a NikonTi-E Eclipse inverted confocal laser-scanning microscope equipped with a 60× Plan Apo VC Oil-immersion DIC N2 objective, a Nikon C2 plus confocal laser scan head and a pinhole of 1.2 airy unit (30 µm). The laser scanning Nikon Confocal Workstation with Galvano scanner, and lasers 408, 488, 561, and 637 nm was used, attached to a Nikon Eclipse Ti inverted microscope. Images were captured in 1024 × 1024 pixels format using NIS-Elements software (Nikon Corporation) and the raw images were processed using ImageJ software. The standardized optical setups of imaging, pin-holes, objective, laser power, and photomultiplier gain were kept constant while recording the data for all measurements.

The identity of the cell line was verified based on microsatellite genotyping by the ECACC Cell Line Identity Verification Service. The STR profiles matched the profiles of the cell lines as deposited in the ATCC and ECACC STR databases.

**Atomic force microscopy**. All AFM images of the trCLN3-L4 and HyApNc aggregates were taken by using a Nanowizard III AFM (JPK Instruments, Berlin) in tapping mode. ACTA probes with silicon tips were used for imaging in dry mode in air. A volume of 3 µL (5 mM) of a solution of magnesium acetate in water was deposited on a freshly cleaved mica surface layer and allowed to incubate for 3 min and afterward the surface was rinsed with 2 × 10 µL of milli-Q water and dried under air pressure. For imaging a volume of 3 µL of the trCLN3-L4/HyApNc solutions in ultra pure water were spotted on the pre-treated mica surface and allowed to incubate for 1 min. After an incubation time of 1 min on the mica surface the excess sample solution was gently shaken off and the mica surface was blown dry with air pressure and mounted to the AFM microscope for immediate imaging. The raw AFM data were processed using the JPK processing software.

**TEM analysis**. The size and structure of the trCLN3-L4 nanoconstructs were analyzed by negative stain electron microscopy. Samples were prepared using negative staining[74]. In brief, carbon coated grids (Quantifoil Micro Tools GmbH, Jena, Germany, 200 mesh) were glow discharged to render the surface hydrophilic prior to applying samples. 10 µL of an aqueous solution of trCLN3-L4 were applied to the grid. Afterward excess solution was carefully blotted off using filter paper followed by three times washing with ddH$_2$O. In the final step, grids were stained with negative staining reagent by placing them (plastic side down) on a 10 µL drop of freshly prepared 2% (v/v) uranyl formate aqueous staining solution. TEM micrographs were recorded using a JEOL JEM 2200 FS electron microscope (JEOL, Japan) operated at 200 kV. The size of the micelles measured on the TEM images could typically be observed in a range between 20 and 25 nm.

**ESI mass spectrometry**. Molecular weights of the trCLN3-L4 and **DxR**-L4 motifs were analyzed by ESI-LCMS in negative ion mode using a Bruker Esquire HCT 6000 ion-trap MS system with an electrospray ionization source in line with an Agilent 1100 series HPLC system with a ZORBAX SB-18 analytical column (2.1 × 50 mm). An elution buffer (10 mM TEA + 100 mM HFIP) in combination with linear gradients of acetonitrile from 0 to 80% in 30 min was used as mobile phase for analysis. The m/z ratio is calculated by deconvolution of the ionic fragments using Bruker Compass Data Analysis Software.

**Serum stability of lipidated and non-lipidated trCLN3 derivatives**. Serum stabilities of trCLN3, its two point mutant non-binding variant trCLN3.mut, and their corresponding lipid-functionalized derivatives trCLN3-L4 and trCLN3.mut-L4 were investigated in FCS and HBS. For this purpose, the aptamer motifs were labeled at their 5′-end with $^{32}$P to form radiolabeled oligonucleotides. The degradation tests of all the aptamer motifs were performed for 0–72 h at 37 °C. 6 pmol (12 µL of 0.5 µM) of the radio-labeled aptamer (5′-end-labeled with γ-$^{32}$P) was incubated in a volume of 300 µL freshly thawed PBS-buffered FCS or HBS (270 µL serum + 30 µL 10 × PBS). For each measurement 10 µL of the samples were

removed, mixed with 90 µL of gel loading buffer (80% formamide + 5 mM EDTA + 0.01% SDS) and subsequently stored at −20 °C. Aliquots of samples were taken after indicated time intervals of 0, 0.3, 1.5, 3, 6, 24, 48, 60, and 72 h, respectively. The serum stability of the aptamer in FCS or in HBS at different time intervals were analyzed on a denaturing PAGE by loading 10 µL of each sample onto a 10% TAE-Urea gel and running the gels for 90 min at 350 V. Gels were wrapped in clingfilm and exposed to a phosphorimager screen in a closed cassette over a period of 12 h and finally the residual intact aptamer bands were analyzed by scanning the screen in a phosphorimage-scanner (FujiFilm FLA 3000). Intensities of the residual intact aptamer bands were calculated applying AIDA image analyzer software program. Serums half-lives of the selected aptamers were determined by using a half-life curve-fitting data analysis program (GraphPad Prism).

**Assembly of trCLN3-L4 and HyApNc nanoconstructs.** The assembly of both the homogeneous nanoconstructs and hybrid micellar nanoconstructs (HyApNc) in aqueous solution, induced by microphase separation, with an outer shell of aptameric DNA and an inner core of the hydrophobic lipids was performed by employing a simple heating and cooling procedure. An aqueous solution of 250 pmol of trCLN3-L4 was added to 250 pmol of DxR-L4 motif dissolved in a volume of 50 µL milli-Q H$_2$O (10 µM solution). The resulting solution was heated to 90 °C for 10 min and subsequently cooled down to a temperature of 10 °C at a rate of 1 °C per 10 min. In case of the aptamers functionalized with fluorescent markers, the solutions were heated up to 70 °C instead of 90 °C and then gradually cooled down to a temperature of 10 °C at a rate of 1 °C per 10 min using a thermocycler.

**Loading HyApNc carrier with DxR.** DxR-loaded hybrid-aptameric nanoconstruct (HyApNc-DxR) was prepared by mixing trCLN3-L4 **3** with DxR-L4 **4** motif in 1:1 ratio with a 10-fold excess of DxR in binding buffer (1× PBS + 1 mM MgCl$_2$). The solution was incubated at 90 °C for 10 min and slowly cooled down to room temperature overnight at a rate of 1 °C per 10 min in order to intercalate DxR into the DxR-L4 motif. The DxR-loaded HyApNc was transferred to an Amicon® Ultra-0.5 centrifugal filter column with 3 K molecular weight cutoff and excess of free DxR which is not intercalated into DxR-L4 motif was removed by three times consecutive centrifugation at 14,000 × g for 10 min at room temperature while adding fresh binding buffer at each centrifugation step. After each centrifugation step, a UV/vis spectrum of the flow through washing was recorded and a reduction in DxR absorbance further confirmed the successive removal of excess DxR through repeated washing.

**Cell viability assay.** To assess the cytotoxicity of free DxR and HyApNc-DxR in NCI-H1838 lung cancer cells, first the H1838 cells (2 × 10$^4$ cells per well) were seeded in a 96-well plate and grown for 24 h. The cells were then washed with 1× PBS (200 µL) and subsequently treated with free DxR (as control), HyApNc-DxR, or HyApNc.mut-DxR in a dose-dependent way with a final DxR concentration ranging from 0.125 to 50 µM per well. After 2 h post treatment, the cells were washed; the RPMI medium was replaced with a fresh RPMI medium, and subsequently either irradiated with UV light for 5 min ($\lambda = 365$ nm; 350 mW cm$^{-2}$), or not irradiated. Afterward the cells were incubated for another 24 h at 37 °C.

For time-dependent cytotoxicity assays, cells were grown at different seeding densities of 10,000, 15,000, 20,000, and 30,000 cells/well in a 96-well plate for 24 h. The cells were then washed with 1× PBS and subsequently incubated with unloaded HyApNc, HyApNc-DxR, and HyApNp$_{w/oAz}$-DxR, respectively, with a final DxR concentration of 8 µM in the culture medium. After 2 h post treatment, the cells were washed; the RPMI medium was replaced with fresh RPMI medium, and subsequently either irradiated with UV light for 5 min ($\lambda = 365$ nm; 350 mW cm$^{-2}$), or not irradiated. Then the cells are allowed to culture for another 8, 24, or 48 h, respectively.

Then for both experiments, 15 µL of an MTT stock solution (5 mg mL$^{-1}$) was added to each well and the cells were incubated at 37 °C for 6 h. After 6 h post treatment with MTT solutions, 100 µL of the SDS-HCL solution was added to each well and mixed thoroughly with a pipette and incubated at 37 °C for an additional 12 h. Finally, the absorbance was measured at $\lambda = 570$ nm by using a Tecan Infinite® M1000 PRO microplate reader.

**Data availability.** The data sets within the article and supplementary files generated during the current study are available from the authors upon request.

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

## Acknowledgements

We acknowledge financial support from the Alexander von Humboldt Foundation and the European Research Council (ERC Advanced Grant). We thank N.S. Hamedani for providing fresh Human Blood Serum, and A. Schmitz, H. Beckert, Y. Aschenbach-Paul, and T. Hornung for help, discussions, and critical reading of the manuscript.

## Author contributions

D.P., V.A. and M.F. conceived and designed the experiments, D.P. and V.A. performed all the chemical/oligo synthesis, confocal imaging, and cell-based experiments. R.Z. performed the FTRET studies and FRET efficiency data analysis. S.I. performed TEM imaging. All authors discussed the results and commented on the manuscripts, D.P. and M.F. wrote the manuscript, and M.F. supervised the project.

## Additional information

**Competing interests:** D.P., V.A., S.I., and M.F declare competing financial interests: a patent application on this work has been submitted and the patent has been licensed by Caris Life Sciences. The remaining author declares no competing financial interests.

