## [Peer Review File · Nature Communications]

Reviewers' comments:

Reviewer #1 (Remarks to the Author):

The manuscript by Prusty et al. describes the design and implementation of an interesting aptamer-based nanoparticle platform for the targeted delivery of therapeutic cargoes to cells. The idea is indeed interesting. Assuming all of the parts are working, couched as a proof of concept study on how one can leverage aptamers, oligonucleotides and chemistry to generate novel self-assembling particles for drug delivery, I like this paper a lot. However, many of the experiments are poorly designed or ill-suited to demonstrate that the system is working the way the authors describe. Of most significant concern is that fact that the while the aptamer used are reported to bind their target at ~40nM, the assays on cells are all done at very high concentrations, 10-40uM (micromolar). These concentrations are not only unrealistic for in vivo experiments, but likely are confounded by non-specific interactions. Surprisingly such interactions do not seem to confound the data/results. However, if the molecules are working as described then the authors should see cell labeling and efficacy at much, much lower concentrations. As a result, although the results are of potential interest to the readers of Nature Communications (as well as the aptamer and targeted delivery fields), I cannot recommend publication without significant additional experimentation and revision.

Major comments:

1. Micelle stability has been shown in serum to be severely affected by the presence of serum proteins which alter the equilibrium, tearing apart the micelle. This is a significant problem for the use of particles in vivo and likely an issue for experiments performed here. As these experiments were performed at 10 uM (a very high and unrealistic concentration) and in 10% FBS (hardly in vivo conditions), the problem may not manifest in the experiments performed here. However, this is a significant and often overlooked issue. The authors should review the literature on this problem (see, for example, Kastantin M, Missirlis D, Black M, Ananthanarayanan B, Peters D, Tirrell M J Phys Chem B. 2010 Oct 7; 114(39):12632-40.). The inclusion of some experiments to address if this is an issue with their system (likely it is) is warranted. Adding a discussion of this problem to the manuscript is also warranted.

2. The other issue with micelle stability in vivo is dilution. Even if the micelles are resistant to the presence of serum proteins, will they remain intact following injection systemically where they will be significantly diluted.

3. These aptamers are reported to bind their target with ~40 nM binding affinity. It is quite puzzling (and troubling) that the microscopy experiments were done at 10 uM. These are exceptionally high concentrations. Did they not see signal at lower concentrations? Based on the multivalent nature of the particles, one would actually expect the binding constants to be much lower than observed with the aptamers themselves.

4. Confocal microscopy is not an appropriate means to demonstrate specificity. In general, microscopy lacks the dynamic range necessary to really provide useful information regarding what is signal and what is background binding. Specificity and apparent affinities should be assessed using alternate analyses such as flow cytometry which will provide a much better picture of what a positive signal and what the background signal (from a mutant aptamers) looks like. Dose response curves for the particles versus the free aptamers on cells would provide a wealth of valuable and interesting information regarding the function of these particles.

5. Throughout the manuscript the authors refer to cytoplasmic staining. This fact is highly unlikely.

cMet is endocytosed via a clatherin mediated mechanism. To this reviewer's knowledge, this does not allow for cytoplasmic access. Few, if any, endocytic routes do. The staining the authors are seeing is likely an artifact of the fixing process. Imaging is best performed using live cells. Or, as noted above, as a follow-up to flow cytometry. The authors should be careful about claims to staining the cytoplasm.

6. The serum stability experiments are interesting, but it's not clear how relevant they are, as stored frozen serum typically lacks the potency of fresh serum. For better impact, these experiment would be better performed using fresh mouse or human serum.

7. It's not clear why the HyAptNc.mut should have poor FRET. There should be no intrinsic difference in the stability of this particle with respect to particles made using the functional aptamer. The only difference would presumably be in the total amount of cell uptake. This fact raises concerns that the FRET analysis is in part an artifact of the analysis and is affected by sample brightness.

8. Cell killing should be performed as a dose response to determine the IC50 for the formulation. As is, the single concentration experiment is not sufficient. Is the construct really on cells at 40 μ M? Is this the concentration of the particle, the aptamer or the DOX? Was cell killing really assessed 8 hours after treatment? This is surprisingly short (see Cancer Res. 2004 Jan 15;64(2):711-8). If DOX is truly killing the cells, a much more pronounced killing would be observed if a longer time period was allowed after initial treatment (24-96hrs).

9. cMet is activated by dimerization. Is it possible the aptamer-particles are crosslinking receptors leading to activation?

Reviewer #2 (Remarks to the Author):

Here Prusty et al. put forward an advanced drug-loading lipid-DNA self-assembled nanoparticle capable of doxorubicin loading and UV/Vis light-switchable conformation changes. Such a system is highly novel and interesting. Overall, I feel the authors do present good controls to show the system is working as described, which is remarkable. I have the following suggestions or questions:

1) In vitro drug release seems lacking and should be presented robustly. This could include:

- release of Dox with and without light irradiation
- in media such as PBS and serum
- ideally with and without light irradiation with a construct lacking the azobenzene.

2) I do not think it is beyond the scope of work to carry out preliminary serum stability and clearance of the drug-loaded construct in mice with IV administration. This is relevant given the remarkable in vitro serum stability shown. It would be interesting to see the correlation between DxR and DNA clearance. Additionally, it will be interesting to see if the particles degrade as per techniques of Fig 2, but using serum sampling. Of course, anti-tumor growth studies would be most welcome as well if possible.

3) Particle uptake is suspected to occur via receptor mediated endocytosis. I am curious if Dox might eventually be released in the endosomes/lysosomes and translate to the nucleus, even without UV light triggering. Perhaps checking Dox localization beyond 2 hrs would be resolve this.

4. The authors could mention some additional relevant background including:

- UV switchable nanoparticles (Tong et al., 2014 PNAS doi:10.1073/pnas.1315336110)

-Light-triggered Dox release (Carter et al., Nat Comm, 2014 doi:10.1038/ncomms4546)
-Aptamer Dox particles (Liu et al., 2016 Biomat, doi: 10.1016/j.biomaterials.2016.03.013)
-Chemophototherapy (Luo et al., Adv Sci 2016 10.1002/advs.201600106)

Reviewer #3 (Remarks to the Author):

In this article the authors claimed that they designed and created an unique aptamer-based nanostructure for use as a targeted drug delivery system. They also claimed that the resulting aptamer-based hybrid nano drug delivery carriers had enhanced serum stability against nuclease, facilitated intracellular drug (doxorubicin) to target cancer cells, and finally controlled drug release in the cells in response to light trigger. The authors provided solid evidence by which they could claim such properties of the aptamer hybrid nanostructure. Most of experiments and data analysis presented herein sound well. It is of course sure that the hybrid nanostructure is a sophisticated system and an interesting delivery vehicle that may attract attention of the readers working in the field of nanomedicine.

However, there are several critical concerns with regard to its potential application for use in vivo or further in clinic.

1. Around 350 nm of UV-Vis light is not proper to be used for cancer therapy in vivo, which is far from near IR range. The present system would work only in the in vitro cell level.
2. There are numerous drug delivery carriers that work very well in the in vitro, but would not work in vivo. Therefore, most of works in this drug delivery field has been demonstrating their feasibility in at least one tumor model in vivo. Without very convinced in vivo anti-tumor efficacy, this work should not be published in the top journal like Nature Communications.

Reviewer 1

Reviewer 1 found our manuscript to be “indeed interesting” and stated that “Assuming all of the parts are working, couched as a proof of concept study on how one can leverage aptamers, oligonucleotides and chemistry to generate novel self-assembling particles for drug delivery, I like this paper a lot”. Reviewer 1 requested major revisions as follows:

1. Micelle stability has been shown in serum to be severely affected by the presence of serum proteins which alter the equilibrium, tearing apart the micelle. This is a significant problem for the use of particles in vivo and likely an issue for experiments performed here. As these experiments were performed at 10 μM (a very high and unrealistic concentration) and in 10% FBS (hardly in vivo conditions), the problem may not manifest in the experiments performed here. However, this is a significant and often overlooked issue. The authors should review the literature on this problem (see, for example, Kastantin M, Missirlis D, Black M, Ananthanarayanan B, Peters D, Tirrell M J Phys Chem B. 2010 Oct 7; 114(39):12632-40.). The inclusion of some experiments to address if this is an issue with their system (likely it is) is warranted. Adding a discussion of this problem to the manuscript is also warranted.

Answer: Point 1 by Reviewer 1 can be considered a summary comment of specific points raised later in the report, namely point 2 (micelle stability upon dilution), point 3 (microscopy done at 10 μM micellar concentration), and point 6 (stability in fresh human serum). We do address all these points in additional experiments. Please see our answers given to the specific points below.

2. The other issue with micelle stability in vivo is dilution. Even if the micelles are resistant to the presence of serum proteins, will they remain intact following injection systemically where they will be significantly diluted.

Answer: We performed additional cell-internalization experiments at concentrations below 10 μM Atto647-labelled HyApNc, namely at 1 μM and 0.2 μM . The data are shown in the new Supp. Fig S14 and are discussed in the main text on p. 10/11. We still see significant internalization at 1 μM (Fig S14b). As expected, when the Atto647N-3 concentration was further reduced to 0.2 μM , which is below the CMC of 0.3-0.35 μM , a significantly weaker fluorescence signal was observed.

3. These aptamers are reported to bind their target with ~40 nM binding affinity. It is quite puzzling (and troubling) that the microscopy experiments were done at 10 μM . These are exceptionally high concentrations. Did they not see signal at lower concentrations? Based on the multivalent nature of the particles, one would actually expect the binding constants to be much lower than observed with the aptamers themselves.

Answer: This point relates to point 2 – see our answer to point 2. If we would have done the experiments at a concentration of lipidated aptamer that is around the 40 nM binding

affinity reported for these aptamers, we would not have formed any micelles, as this concentration is roughly four orders of magnitude below the CMC measured in Supporting Information Chapter 2. Indeed, even at 0.2 μM , we already see markedly reduced internalization compared to 1.0 μM – clearly indicating that we need to perform the internalization experiments at concentrations above the CMC of 0.3-0.35 μM .

4. Confocal microscopy is not an appropriate means to demonstrate specificity. In general, microscopy lacks the dynamic range necessary to really provide useful information regarding what is signal and what is background binding. Specificity and apparent affinities should be assessed using alternate analyses such as flow cytometry which will provide a much better picture of what a positive signal and what the background signal (from a mutant aptamers) looks like. Dose response curves for the particles versus the free aptamers on cells would provide a wealth of valuable and interesting information regarding the function of these particles

Answer: We have included the suggested Flow Cytometry experiments (see SI Chapter 10 for the Methods) and report them in the new Figure 5b for the Atto-labeled nanoconstructs, and in the new Fig. 7b for the doxorubicin-release under various conditions in direct comparison with the confocal microscopy results. The results of Fig. 5b are described in the text on p. 11, the results of Fig. 7b are described on p. 13 /14 in the main text.

5. Throughout the manuscript the authors refer to cytoplasmic staining. This fact is highly unlikely. cMet is endocytosed via a clatherin mediated mechanism. To this reviewer's knowledge, this does not allow for cytoplasmic access. Few, if any, endocytic routes do. The staining the authors are seeing is likely an artifact of the fixing process. Imaging is best performed using live cells. Or, as noted above, as a follow-up to flow cytometry. The authors should be careful about claims to staining the cytoplasm.

Answer: We no longer refer to cytoplasmic staining. Indeed at lower micelle concentrations it becomes evident that the nanoconstructs are internalized via an endocytic mechanism (Supp. Fig S14b) and are present in endosomes rather than located in the cytoplasm. The fact that we do not see any significant uptake at 4 °C (Fig. 5) further supports this model.

6. The serum stability experiments are interesting, but it's not clear how relevant they are, as stored frozen serum typically lacks the potency of fresh serum. For better impact, these experiment would be better performed using fresh mouse or human serum.

Answer: We performed another series of experiments quantifying the stability of the nanocarriers in fresh human blood serum (see new Fig. 2b, c). The results are discussed on p. 6/7. Clearly the data confirm the increased stability that we had reported for FCS in the initial version of the manuscript.

7. It's not clear why the HyAptNc.mut should have poor FRET. There should be no intrinsic difference in the stability of this particle with respect to particles made using the functional aptamer. The only difference would presumably be in the total amount of cell uptake. This fact raises concerns that the FRET analysis is in part an artifact of the analysis and is affected by sample brightness.

Answer: The HyAptNc-mut nanoparticles lead to a very low FRET signal inside cells because of their marginal uptake. However, reviewer 1 is right when saying that there should not be an intrinsic difference in the stability of particles compared to the functional aptamer. We therefore now include a measurement of the intrinsic fluorescence of the HyAptNc.mut nanoparticles (see new Suppl. Fig. S13, sample F7) that shows similar intrinsic FRET efficiency of HyAptNc.mut compared to HyAptNc.

8. Cell killing should be performed as a dose response to determine the IC50 for the formulation. As is, the single concentration experiment is not sufficient. Is the construct really on cells at 40 μM ? Is this the concentration of the particle, the aptamer or the DOX? Was cell killing really assessed 8 hours after treatment? This is surprisingly short (see Cancer Res.

2004 Jan 15;64(2):711-8). If DOX is truly killing the cells, a much more pronounced killing would be observed if a longer time period was allowed after initial treatment (24-96hrs).

Answer: We appreciate the point made by reviewer 1 (see also point 3 made by reviewer 2). To address this point, we first determined how many molecules of DxR are bound to motif 4 (see new Suppl. Fig. S11). We find 8 molecules of DxR bound to each motif 4. We also state now more clearly that the effective concentration of free DxR is 40 μM both in free form and when intercalated into motif 4 ($5 \mu\text{M}$ nanoconstruct \times 8 DxR = 40 μM DxR_{eff}) on p. 13, top. We then determined IC50 values of cell viability (see new Methods section, main manuscript p. 22) using free DxR and compared these to similar effective DxR concentration when bound to HyApNc or HyApNc.mut with and without UV (see new Fig. 8a). We also determined cell viability at time-points 8h, 24h, 48 h as requested (see new Fig. 8b, c). In the main text these results are described on p. 14-16.

9. cMet is activated by dimerization. Is it possible the aptamer-particles are crosslinking receptors leading to activation?

Although at this point we cannot completely exclude this possibility, we think that activation by dimerization induced by two neighboring aptamers on the same micelle, each of which binds to a separate monomer leading to a functional dimer is unlikely. Receptor dimerization leading to receptor activation is a highly delicate and tightly regulated process.

Reviewer 2

Reviewer 2 considered our approach to be highly novel and interesting and had the following suggestions and questions:

1. In vitro drug release seems lacking and should be presented robustly. This could include:
 - release of Dox with and without light irradiation
 - in media such as PBS and serum
 - ideally with and without light irradiation with a construct lacking the azobenzene.

Answer: We appreciate the point made by reviewer 2. The DxR-release experiments are shown in the new Supplementary Figure S12 and discussed in the main text on p. 8/9. We also synthesized the construct lacking azobenzene as suggested, and used it to assemble HyApNc_{w/o} AZ-DxR. This nanocarrier could not be triggered to release DxR due to it lacking the azobenzene moieties. We used it together with the HyApNc-DxR and free DxR to measure drug release in cells by confocal microscopy and by flow cytometry (see new Figure 7a,b). The data are discussed on p. 13/14 in the main text.

2. I do not think it is beyond the scope of work to carry out preliminary serum stability and clearance of the drug-loaded construct in mice with IV administration. This is relevant given the remarkable in vitro serum stability shown. It would be interesting to see the correlation between DxR and DNA clearance. Additionally, it will be interesting to see if the particles degrade as per techniques of Fig 2, but using serum sampling. Of course, anti-tumor growth studies would be most welcome as well if possible.

Answer: We performed additional serum stability studies in vitro (see response to reviewer 1's point 6). We did not do anti-tumour growth studies in agreement with the editor.

3. Particle uptake is suspected to occur via receptor mediated endocytosis. I am curious if Dox might eventually be released in the endosomes/lysosomes and translate to the nucleus, even without UV light triggering. Perhaps checking Dox localization beyond 2 hrs would resolve this.

Answer: We did the experiment (see response to reviewer 1's point 8).

4. The authors could mention some additional relevant background including:
- UV switchable nanoparticles (Tong et al., 2014 PNAS doi:10.1073/pnas.1315336110)
 - Light-triggered Dox release (Carter et al., Nat Comm, 2014 doi:10.1038/ncomms4546)
 - Aptamer Dox particles (Liu et al., 2016 Biomat, doi: 10.1016/j.biomaterials.2016.03.013)
 - Chemophototherapy (Luo et al., Adv Sci 2016 10.1002/adv.201600106)

Answer: We now cite these papers

Reviewer 3

1. Around 350 nm of UV-Vis light is not proper to be used for cancer therapy in vivo, which is far from near IR range. The present system would work only in the in vitro cell level.

Answer: Our manuscript should be considered a proof-of-concept study (as reviewer 1 correctly stated in his/her introductory statement). We mention the drawback that reviewer 3 refers to on p. 16/17 of the manuscript, citing reference 70 as a potential solution to the problem in future studies.

2. There are numerous drug delivery carriers that work very well in the in vitro, but would not work in vivo. Therefore, most of works in this drug delivery field has been demonstrating their feasibility in at least one tumor model in vivo. Without very convinced in vivo anti-tumor efficacy, this work should not be published in the top journal like Nature Communications.

Answer: We did not do anti-tumour growth studies in agreement with the editor.

Changes not requested by reviewers

We added a scheme to Supporting Figure S3 that should make it easier for the reader to comprehend the assembly of the various nanoparticles used in our study. We streamlined the abstract and some parts in the main text to improve readability. All changes to text of the revised version compared to the previous version are marked in the attached pdf file "Prusty et al Edits" that we submit as additional information for reviewing purposes only.

We are grateful to the reviewers for their comments and the time spent on evaluating our manuscript. We found their input to be very valuable. We also thank you for your consideration and for your efforts. We hope the manuscript is now ready for being accepted and look forward to hearing from you.

Reviewers' comments:

Reviewer #1 (Remarks to the Author):

The manuscript by Prusty et al has addressed many of the concerns initially made by this as well as the other reviewers. In particular, additional experiments, including the use of flow cytometry, have been performed to confirm the specificity of uptake. Additionally, the role of UV excitation in enhancing drug release has been better demonstrated with added studies.

This remains a nice manuscript, and a nice proof of concept study which will be of interest to others in the field. The work is well done and suitable for publication in NComm, however, I still have concerns regarding the work which will require some additional revision. These points are outlined below:

1. In my original review I noted concerns regarding micelle stability which I'm not sure the authors have sufficiently answered. Micelle stability is an issue on multiple levels. The first is the CMC. The authors demonstrate that CMC is $\sim 0.3 \mu\text{M}$. Good. However, what happens when these get injected into an animal where they will be significantly diluted? By the authors own accord, significant staining is only achieved when high concentrations are used, concentrations above the CMC. Yes, I recognize that this is a proof of concept study, but this is going to be a big problem moving this technology forward into animals and the limitation should be added to the Discussion.

Second, what happens to these particles in complex media or serum where serum proteins can bind lipids, skewing the equilibrium and rip the particles apart? This is a significant and important problem with micelles that is often overlooked (or ignored). The authors should refer to work by the Tirell Group (for example J Phys Chem B. 2010 Oct 7;114(39):12632-40.) as well as others who have worked to solve this problem (see for example work from the Xu group, J. Am. Chem. Soc., 2012, 134 (28), pp 11807-11814). This really should be addressed experimentally in the manuscript. If the particles fall apart targeting will not work except in culture where protein content is low. In fact, I would guess that these particles will not stay intact in serum, and dissociate very quickly. That's a problem and the authors really need to test this and address this limitation as well as potential solutions in the Discussion.

2. The authors state concern that "In the case of aptamer-drug conjugates, covalent linking of targeting units to cytotoxic agents is further limited by the concern that the attachment may alter their biological activity." I must have missed this sentence the first time. Do the authors really think a drug conjugate is going to be more problematic than engineering the targeting to self-associate into an aggregate? I disagree and think the authors are overstating their case.

3. It's quite surprising that with a reported K_d of $\sim 50 \text{nM}$ that the labeled aptamer on its own does not efficiently stain cells. It's quite nice that the authors can, in essence, rescue this using their approach. In this respect it would be nice for the authors to highlight this fact to a) call attention to that fact that protein target binding in solution does not equate to cell surface binding and b) that their approach has the ability to rescue/improve aptamers with poor function.

4. I'm confused by the rationale for doing the experiment in Figure 7b. Whether the DxR is intercalated in the particle or intercalated in the nucleus, the cytometry can't tell. There should essentially be no change with or without UV, and there isn't. It's kind of a silly experiment. The experiment which should be included here (or elsewhere) should be to look at the effect of DxR uptake using a targeted particle or a non-targeted particle.

5. The flow cytometry methods should be included in the manuscript proper.

Reviewer #2 (Remarks to the Author):

The authors have revised the manuscript, however the response left a couple of issues hanging:

1) In the newly added data in Figure S12, the authors demonstrate that their method for quantifying Dox is potentially not reliable. The HPLC elution peak is strikingly broad and this is not discussed. Furthermore, it is not clear how exactly the authors were measuring Dox release from the construct. If they are simply injecting the construct with or without UV irradiation, the total Dox should be the same regardless of whether it is released or not. Light-induced doxorubicin release should be better quantified and methods explained more clearly

2) Perhaps I did not clearly state my second request, which was "I do not think it is beyond the scope of work to carry out preliminary serum stability and clearance of the drug-loaded construct in mice with IV administration." This was not requesting anti-tumour growth studies, as the authors replied. This would be to assess the serum half-life and stability of the Dox (can do with HPLC once proper method is used) and the nucleic acids (similar to the data in Fig 2), which could provide both stability and circulation information.

Please find below the point-by-point response that addresses the points made by reviewers 1 and 2. All changes with respect to the previous version of the manuscript are highlighted in yellow.

Reviewer #1 (Remarks to the Author):

The manuscript by Prusty et al has addressed my of the concerns initially made by this as well as the other reviewers. In particular, additional experiments, including the use of flow cytometry, have been performed to confirm the specificity of uptake. Additionally, the role of UV excitation in enhancing drug release has been better demonstrated with added studies.

This remains a nice manuscript, and a nice proof of concept study, which will be of interest to others in the field. The work is well done and suitable for publication in NComm, however, I still have concerns regarding the work, which will require some additional revision. These points are outlined below:

1. In my original review I noted concerns regarding micelle stability, which I'm not sure the authors have sufficiently answered. Micelle stability is an issue on multiple levels. The first is the CMC. The authors demonstrate that CMC is $\sim 0.3\mu\text{M}$. Good. However, what happens when these get injected into an animal where they will be significantly diluted? By the authors own accord, significant staining is only achieved when high concentrations are used, concentrations above the CMC. Yes, I recognize that this is a proof of concept study, but this is going to be a big problem moving this technology forward into animals and the limitation should be added to the Discussion.

Answer: Limitations discussed in Discussion part of the main manuscript page 18/19: "This concept might be further limited..."

Second, what happens to these particles in complex media or serum where serum proteins can bind lipids, skewing the equilibrium and rip the particles apart? This is a significant and important problem with micelles that is often overlooked (or ignored). The authors should refer to work by the Tirell Group (for example J Phys Chem B. 2010 Oct 7; 114(39): 12632-40.) as well as others who have worked to solve this problem (see for example work from the Xu group, J. Am. Chem. Soc., 2012, 134 (28), pp 11807–11814). This really should be addressed experimentally in the manuscript. If the particles fall apart targeting will not work except in culture where protein content is low. In fact, I would guess that these particles will not stay intact in serum, and dissociate very quickly. That's a problem and the authors really need to test this and address this limitation as well as potential solutions in the Discussion.

Answer: As suggested, we tested the stability of the micellar nanoconstruct in 95% human blood serum and, as a control, in 1 mM BSA solution at different time periods

using time-resolved FRET. These new data are described in the main manuscript on page 10/11 (“For efficient cell internalization...”) and supplementary information chapter 9 and Fig. S14).

2. The authors state concern that “In the case of aptamer-drug conjugates, covalent linking of targeting units to cytotoxic agents is further limited by the concern that the attachment may alter their biological activity.” I must have missed this sentence the first time. Do the authors really think a drug conjugate is going to be more problematic than engineering the targeting to self-associate into an aggregate? I disagree and think the authors are overstating their case.

Answer: We toned down the statement in the introduction part of the manuscript (p. 2, bottom: “In the case of...”).

3. It's quite surprising that with a reported K_d of ~50nM that the labeled aptamer on its own does not efficiently stain cells. It's quite nice that the authors can, in essence, rescue this using their approach. In this respect it would be nice for the authors to highlight this fact to a) call attention to that fact that protein target binding in solution does not equate to cell surface binding and b) that their approach has the ability to rescue/improve aptamer with poor function.

Answer: Discussion added in main manuscript page 12, §2: “This result may indicate...”.

4. I'm confused by the rationale for doing the experiment in Figure 7b. Whether the DxR is intercalated in the particle or intercalated in the nucleus, the cytometry can't tell. There should essentially be no change with or without UV, and there isn't. It's kind of a silly experiment. The experiment which should be included here (or elsewhere) should be to look at the effect of DxR uptake using a targeted particle or a non-targeted particles.

Answer: We had mentioned in the previous revision that the additional flow cytometry experiment in Figure 7b does not distinguish between the nuclear accumulation or endoplasmic accumulation, however, the FACS data showed a small shift in overall fluorescence signal due to release of DxR from the intercalated motif (only in case of particle having DMAB modification and exposed to UV irradiation). This supports the results from the confocal cell studies, which indicate a nuclear accumulation only in case of motifs having DMAB modifications intercalating to Doxorubicin (Visible from 7a II-V). As suggested by the reviewer, the FACS data for DxR uptake using a targeted particle (HyApNc-DxR) versus the mutated non-targeted (HyApNc.mut-DxR) particles were included in the manuscript Figure 7b, I. (description main manuscript p. 14 bottom: “As a control, the DXR uptake...”, and p. 15: “moreover, cells incubated with HyApNc-DXR...”).

5. The flow cytometry methods should be included in the manuscript proper.

Answer: methods described in Supporting Information Chapter 11.

Reviewer #2 (Remarks to the Author):

1) In the newly added data in Figure S12, the authors demonstrate that their method for quantifying Dox is potentially not reliable. The HPLC elution peak is strikingly broad and this is not discussed. Furthermore, it is not clear how exactly the authors were measuring Dox release from the construct. If they are simply injecting the construct with or without UV irradiation, the total Dox should be the same regardless of whether it is released or not. Light-induced doxorubicin release should be better quantified and methods explained more clearly.

Answer: It is known that the 1:1 phenol/ chloroform mixture can remove unbound excess doxorubicin from the duplex DNA motif (see: Stuart C.H., et al. *Bioconjugate Chem.* **2014**, 25: 406-13) in the presence of DNA duplex without removing intercalated DxR. Therefore before each injection into HPLC, the constructs after irradiation either for the release study or for the time dependent thermal release study, were first washed with 1:1 phenol/ chloroform mixture and chloroform and then injected into HPLC (we already mentioned this in the main manuscript, p. 8 and described the detailed procedure in the supporting information chapter 7 and included the above reference).

The broad peak might result from the interaction of the lipidated motif 4 with the solid phase of the HPLC column. Lipid modifications can broaden the peak significantly. Note that we performed the HPLC analysis using an ion exchange column.

2) Perhaps I did not clearly state my second request, which was "I do not think it is beyond the scope of work to carry out preliminary serum stability and clearance of the drug-loaded construct in mice with IV administration." This was not requesting anti-tumour growth studies, as the authors replied. This would be to assess the serum half-life and stability of the Dox (can do with HPLC once proper method is used) and the nucleic acids (similar to the data in Fig 2), which could provide both stability and circulation information.

Answer: We have now added new experiments, in which we determined the stability of the micellar nanoconstructs in human blood serum as well as in presence of 1 mM solution of BSA at 37 °C (see answer to reviewer 1, first request, second part). The data indicate a half-life of 14 h in HBS, showing that the experiments presented in this work provide a reasonable model for in vivo conditions as far as micelle breakup is concerned (described in main manuscript page no- 10-11 and supplementary information chapter 9 and Fig S14).

We appreciate the point made by reviewer 2. We cannot exclude that the observed half-life is further reduced when testing HyApNc in a mouse model. Unfortunately, we are not equipped to perform the requested pharmacokinetic experiments in an animal model. However, we now state on p. 11, end of first §: "It is possible that the $t_{1/2}$ of HyApNc will be further reduced in the blood-stream in vivo. However, if necessary for in vivo applications the half-life of HyApNc could be further increased by elongating the lipid chains and/or by using unsaturated lipids and crosslinking them at the core of the nanostructures".

Again we thank the reviewers for their comments and the time spent on evaluating our manuscript.

REVIEWERS' COMMENTS:

Reviewer #1 (Remarks to the Author):

The authors have done a nice job addressing this reviewers concerns.

Reviewer #2 (Remarks to the Author):

I am satisfied with the revised manuscript.